# A muscle-epidermis-glia signaling axis sustains synaptic specificity during allometric growth in *Caenorhabditis elegans*

Jiale Fan[1†], Tingting Ji[1†], Kai Wang[1†], Jichang Huang[2], Mengqing Wang[1], Laura Manning[3], Xiaohua Dong[1], Yanjun Shi[1], Xumin Zhang[2], Zhiyong Shao[1]*, Daniel A Colón-Ramos[3,4]*

[1]Department of Neurosurgery, the State Key Laboratory of Medical Neurobiology and MOE Frontiers Center for Brain Science, the Institutes of Brain Science, and Zhongshan Hospital, Fudan University Shanghai, Shanghai, China; [2]State Key Laboratory of Genetic Engineering, Department of Biochemistry, School of Life Sciences, Fudan University, Shanghai, China; [3]Program in Cellular Neuroscience, Neurodegeneration and Repair, Department of Neuroscience and Department of Cell Biology, Yale University School of Medicine, New Haven, United States; [4]Instituto de Neurobiología, Recinto de Ciencias Médicas, Universidad de Puerto Rico, San Juan, Puerto Rico

*For correspondence:
shaozy@fudan.edu.cn (ZS);
daniel.colon-ramos@yale.edu
(DACó-R)

[†]These authors contributed
equally to this work

Competing interests: The
authors declare that no
competing interests exist.

Reviewing editor:  Oliver
Hobert, Howard Hughes Medical
Institute, Columbia University,
United States

**Abstract** Synaptic positions underlie precise circuit connectivity. Synaptic positions can be established during embryogenesis and sustained during growth. The mechanisms that sustain synaptic specificity during allometric growth are largely unknown. We performed forward genetic screens in *C. elegans* for regulators of this process and identified *mig-17*, a conserved ADAMTS metalloprotease. Proteomic mass spectrometry, cell biological and genetic studies demonstrate that MIG-17 is secreted from cells like muscles to regulate basement membrane proteins. In the nematode brain, the basement membrane does not directly contact synapses. Instead, muscle-derived basement membrane coats one side of the glia, while glia contact synapses on their other side. MIG-17 modifies the muscle-derived basement membrane to modulate epidermal-glial crosstalk and sustain glia location and morphology during growth. Glia position in turn sustains the synaptic pattern established during embryogenesis. Our findings uncover a muscle-epidermis-glia signaling axis that sustains synaptic specificity during the organism's allometric growth.

## Introduction

Proper nervous system architecture depends on establishing and maintaining precise connectivity between pre- and post-synaptic partners. Failure to maintain proper synaptic connectivity leads to impaired nervous system function and neurological disorders (*Mariano et al., 2018*). Remarkably, circuit architecture is largely maintained during growth even as tissues change in relative size and position to each other. The mechanisms that sustain synaptic connectivity during growth remain largely unknown.

Our understanding of correct synaptic connectivity primarily derives from developmental studies examining the precise positioning of synapses during their biogenesis (*Kurshan and Shen, 2019*; *Park et al., 2018*; *Rawson et al., 2017*). These studies indicate that precise connectivity during development occurs through orchestrated signaling across multiple tissues. While cell-cell

recognition and signaling between synaptic partners are pivotal for synaptogenesis, non-neuronal cells are also critical in vivo to guide synaptic specificity (*Colón-Ramos, 2009*; *Margeta and Shen, 2010*; *Sanes and Yamagata, 2009*; *Shimozono et al., 2019*). For example, during development, guidepost cells such as glia instruct synaptic specificity by secreting positional cues to the extracellular matrix (ECM) (*Ango et al., 2008*; *Colón-Ramos et al., 2007*; *Eroglu and Barres, 2010*; *Molofsky et al., 2014*; *Shen and Bargmann, 2003*; *Tsai et al., 2012*; *Ullian et al., 2001*). Therefore, non-cell autonomous mechanisms, mediated through the ECM, can coordinate synaptic connectivity during development in vivo.

Less is known about the factors required for sustaining the synaptic pattern during post-embryonic growth. Multiple studies have identified mechanisms required for post-embryonic maintenance of synapses, but not synaptic positions. These studies on post-embryonic maintenance of synapses have resulted in the discovery of important regulators of synaptic stability, density and morphology (*Burden et al., 2018*; *Cherra and Jin, 2016*; *Hasan and Singh, 2019*; *Lin and Koleske, 2010*; *Luo et al., 2014*; *Sytnyk et al., 2017*), including roles for ECM components in the maintenance of synapses of both the peripheral and the central nervous system. In the peripheral nervous system (PNS), disrupting ADAMTS metalloproteases and basement membrane proteins impairs the post-embryonic maintenance of the morphology of neuron-muscle synapses (called neuromuscular junctions, or NMJs) (*Cescon et al., 2018*; *Dear et al., 2016*; *Heikkinen et al., 2019*; *Kurshan et al., 2014*; *Qin et al., 2014*; *Singhal and Martin, 2011*). Basement membrane proteins are also important for neuron-neuron synapses in the central nervous system (CNS) (*Heikkinen et al., 2014*). However, unlike NMJs in the PNS, most neuron-neuron synapses in the CNS are not in direct contact with the basement membrane (*Heikkinen et al., 2014*; *Krishnaswamy et al., 2019*). How the basement membrane sustains CNS neuron-neuron synapses, particularly during brain allometric growth, remains unknown.

Sustaining the relative synaptic positions during growth, and therefore embryonically derived synaptic specificity, is important for sustaining circuit integrity. As an animal grows, organs scale in different proportions relative to body size. This conserved principle is termed 'allometry' (*Huxley, 1924*; *Huxley, 1936*). For relevance to the brain, neocortical white matter and grey matter scale differently from each other, indicating that specific sub-structures of the brain scale allometrically to total brain size (*de Jong et al., 2017*). Presynaptic partners, postsynaptic partners and non-neuronal cells that provide positional cues also scale allometrically during growth. We do not know the underlying mechanisms that sustain embryonically-derived circuit architecture as different tissues disproportionately grow in size.

The nematode *C. elegans* provides a tractable genetic model to examine questions related to sustaining synaptic specificity during growth (*Shao et al., 2013*). After hatching from its egg, *C. elegans* grows an order of magnitude in length during post-embryonic growth (*Knight et al., 2002*). The architecture of the nervous system, which is established during embryogenesis, is largely preserved during this process (*Bénard and Hobert, 2009*). The use of cell-specific promoters in conjunction with in vivo probes permits visualizing and tracking synapses in single neurons of known identity during the lifetime of the organism (*Colón-Ramos et al., 2007*; *Nonet, 1999*).

In our prior work, we identified *cima-1* as a gene required for sustaining the synaptic pattern during growth (*Shao et al., 2013*). In *cima-1* mutants, synaptic contacts are correctly established during embryogenesis, but ectopic pre-synaptic sites emerge as the animals grow. *cima-1* encodes a novel solute carrier in the SLC17 family of transporters that includes Sialin, a protein that when mutated in humans produces neurological disorders (*Verheijen et al., 1999*). However, *cima-1* does not function in neurons. Instead, it functions in nearby epidermal cells to antagonize the FGF Receptor, likely by inhibiting its role in epidermal-glia adhesion (*Figure 1*). Thus, *cima-1* functions in non-neuronal cells during post-embryonic growth to preserve the synaptic pattern (*Shao et al., 2013*).

To further determine the cellular and molecular mechanisms that regulate the synaptic pattern during growth, we performed suppressor forward genetic screens in the *cima-1* mutant background, and identified *mig-17*, encoding a secreted ADAMTS metalloprotease (*Nishiwaki et al., 2000*). We find that the secreted *mig-17* modulates muscle-derived basement membrane proteins. The synapses examined in this study are not in direct contact with the basement membrane. Instead, the basement membrane coats the side of glia facing the pseudocoleum, while glia contact synapses on their other side facing the nerve ring. We find that MIG-17 modifies the muscle-derived basement membrane to modulate epidermal-glial crosstalk and sustain glia location and morphology during

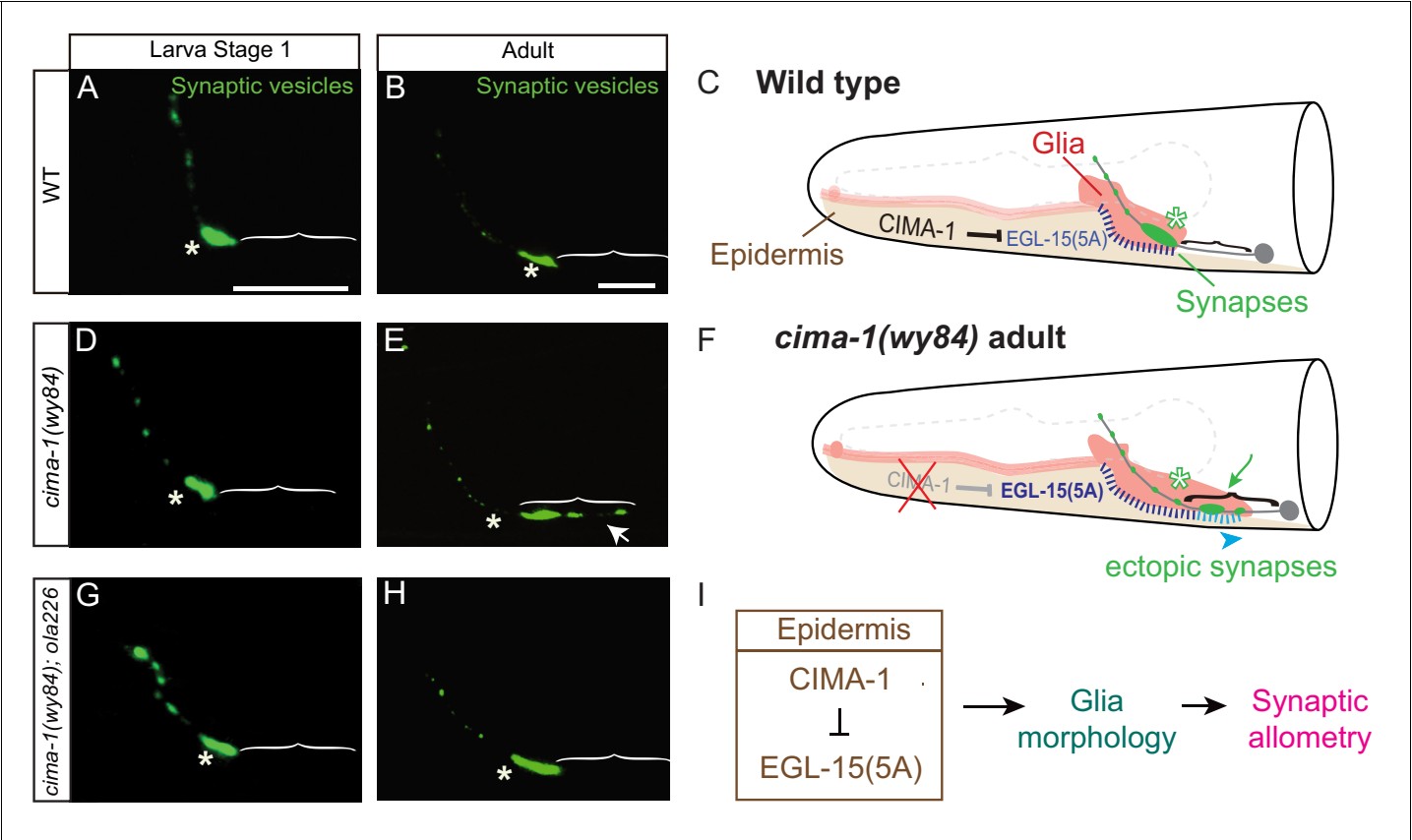

**Figure 1.** Synaptic allometry in AIY neurons. (A–C) Distribution of AIY synapses in wild-type animals, and model. (A–B) Confocal micrograph images of AIY presynaptic sites labeled with the synaptic vesicle marker mCherry::RAB-3 (pseudo-colored green) in wild-type larval stage 1 (L1) animals (A) and adult animals (B). Note that although animals grow (scale bars in A and B both correspond to 10 μm), in wild-type animals the synaptic pattern is sustained from L1 to adults. Asterisks indicate the synaptic-rich Zone 2 and brackets indicate the asynaptic Zone 1 regions of AIY (see *Figure 2A*). (C) Graphical abstract of the findings of *Shao et al. (2013)*. In wild-type animals, CIMA-1 acts in epidermal cells to suppress the epidermally derived FGF Receptor/EGL-15, which in turn maintains VCSC glia morphology, which likely mediates adhesion between the epidermal cell and glia. In cartoon, epidermal cells in beige, glia in red, AIY neuron in grey, synapses in green, Zone 2 region indicated by asterisk and stitches represent contact sites between the epidermis and glia. Also outlined in grey dashed lines, the position of the pharynx for reference. (D–F) As (A–C), but for *cima-1(wy84)* loss-of-function mutants. In *cima-1* loss-of-function mutants, EGL-15(5A)/FGF Receptor protein levels are upregulated, and this promotes adhesion of epidermis to glia and causes glia position and morphology defects during growth (F). This in turn extends the glia-AIY contact site to the asynaptic Zone 1 region, causing ectopic synapse formation in Zone 1 (see also *Figure 1—figure supplement 1C–F*). Blue arrow in (F) represent the changes in glia position and morphology due to increased interaction with epidermal cells, and green arrow marks ectopic synapses in Zone 1 (brackets). (G–H) As in (A–B), but in *cima-1(wy84);ola226* double mutants. Note that the *cima-1* synaptic phenotype (E) is suppressed in the *cima-1(wy84);ola226* double mutant (H). (I) Schematic model of the multi-tissue CIMA-1 regulation of synaptic allometry in AIY. The scale bars in (A) apply to (D and G), and scale bars in (B) apply to (E and H). Both are 10 μm.

The online version of this article includes the following figure supplement(s) for figure 1:

**Figure supplement 1.** Model of CIMA-1 site of action.

growth. Glia location and morphology in turn sustains the presynaptic pattern as the animal grows. Therefore a muscle-epidermis-glia signaling axis, modulated by *mig-17* and the basement membrane, regulates synaptic allometry during growth.

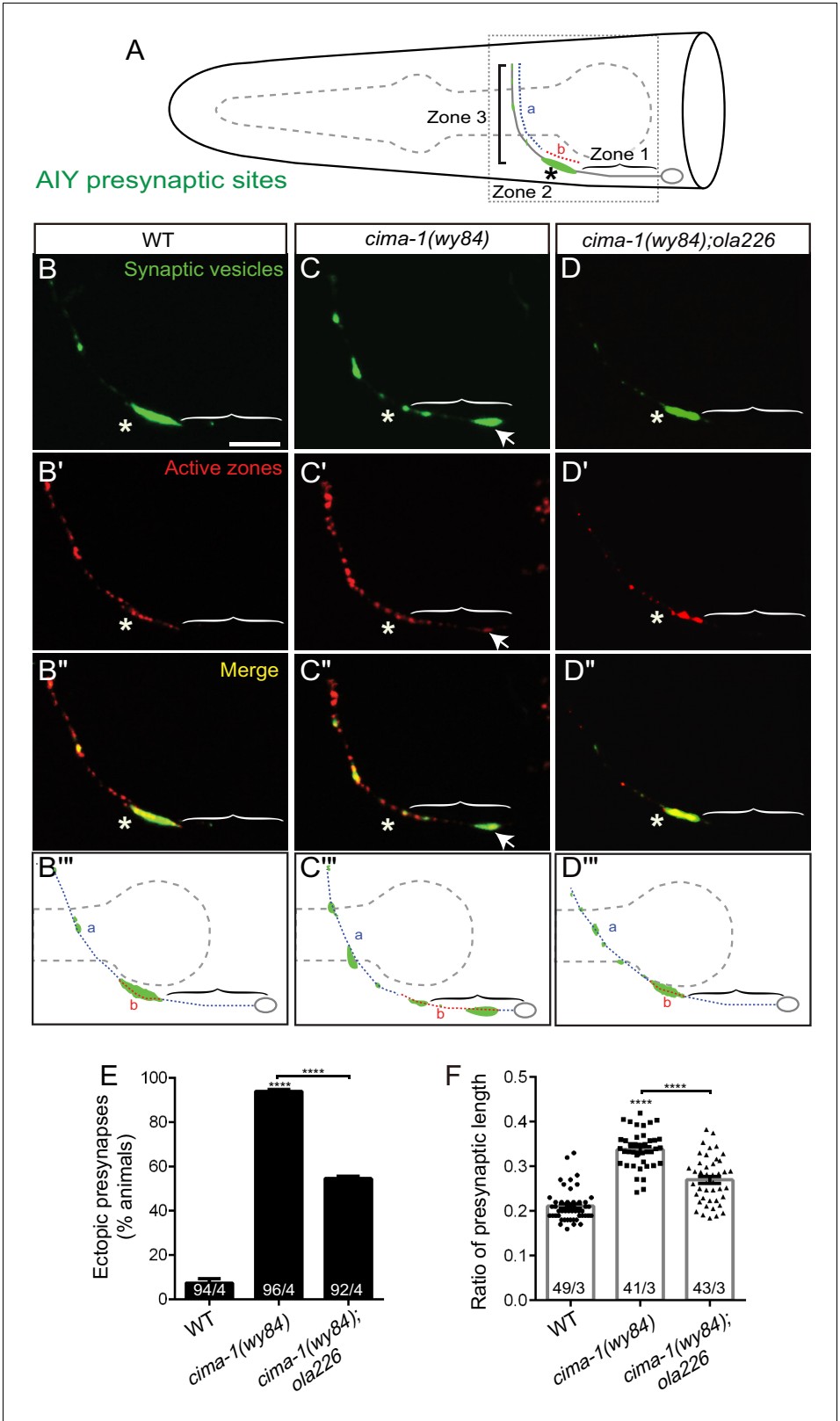

**Figure 2.** Mutant allele *ola226* suppresses *cima-1 (wy84)* synaptic allometry defects in AIY. (**A**) Cartoon diagram of the distribution of presynaptic sites in the AIY interneurons of the nematode *C. elegans*. The head of *C. elegans* (solid black lines) and the pharynx (dashed grey line) are outlined. A single AIY interneuron is depicted in gray, an oval represents the cell body and a solid gray line represents the neurite. Presynaptic puncta are green. The AIY

*Figure 2 continued on next page*

*Figure 2 continued*

neurites can be subdivided into three zones: an asynaptic region proximal to the cell body called Zone 1, a synapse-rich region called Zone 2 (asterisk) and a region with sparse synapses, called Zone 3. The red (b) and blue (a) dashed lines represent synaptic distribution and correspond to Zone 2 and 3 (respectively) in wild-type animals. The dotted box represents the region of the head imaged in B-D'. (**B–D''**) Confocal micrograph images of AIY presynaptic sites labeled with the synaptic vesicle marker mCherry::RAB-3 (pseudo-colored green, **B–D**) and active zone protein GFP::SYD-1 (pseudo-colored red, (**B'–D'**) for wild type (**B, B', B''**), *cima-1(wy84)* mutants (**C, C', C''**) or *cima-1(wy84);ola226* (**D, D', D''**). Merged images display co-localization of synaptic vesicle marker mCherry::RAB-3 and active zone protein GFP::SYD-1 in (**B''–D''**). Schematic diagrams of the observations are depicted in (**B'''–D'''**). Scale bar in (**B**) applies to all images, 10 μm. Asterisk: Zone 2 region; Arrows: ectopic synapses in Zone 1 region (see also *Figure 1—figure supplement 1C–F*). (**E**) Quantification of the percentage of animals displaying ectopic AIY presynaptic sites in the Zone 1 region for indicated genotypes. (**F**) Quantification of the ratio of ventral synaptic length (see red (b) to total synaptic region (sum of the length of blue (a) and red (b) in schematic in (**A and B'''–D'''**)). The total number of animals (N) and the number of times scored (n) are indicated in each bar for each genotype as N/n. Error bars represent SEM. Statistical analyses are based on one-way ANOVA by Tukey's multiple comparison test, ****$p < 0.0001$ as compared to wild type (if on top of bar graph), unless brackets are used between two compared genotypes.

The online version of this article includes the following figure supplement(s) for figure 2:

**Figure supplement 1.** Relationship between body size and synaptic allometry in *cima-1(wy84)* mutants and the *ola226* allele.

# Results

## Mutant allele *ola226* suppresses synaptic allometry defects in *cima-1 (wy84)*

AIY interneurons are a pair of bilaterally symmetric neurons in the *C. elegans* nerve ring. AIYs display a stereotyped and specific pattern of presynaptic specializations (*Colón-Ramos et al., 2007*; *White et al., 1986*). This pattern is established during embryogenesis. Even though animals grow an order of magnitude in length from early embryogenesis to adulthood (from ~100 μm to ~1 mm) (*Knight et al., 2002*; *Shibata et al., 2016*), the AIY synaptic pattern is sustained during growth (*Figure 1A–C* and *Shao et al., 2013*). Here, we term this process of sustaining the synaptic pattern during growth 'synaptic allometry'. Synaptic allometry requires coordination between different tissues to sustain the relative pre- and postsynaptic positions during growth (*Shao et al., 2013*). Which cell types are required, and how they signal to coordinately sustain synaptic allometry is not well understood.

Using forward genetic screens, we previously identified *cima-1* as a gene required for sustaining the synaptic pattern during growth (*Shao et al., 2013*). In *cima-1(wy84)* mutants, the embryonic AIY synaptic pattern developed correctly (*Figure 1D*). However, during growth, synaptic positions were disrupted and ectopic presynaptic sites emerged in the Zone 1 region, a normally asynaptic region of the AIY neuron (*Figure 1E–F* and *Shao et al., 2013*). *cima-1* encodes a solute carrier transporter required in epidermal cells to antagonize the FGF receptor and likely modulate epidermal-glia adhesion (*Shao et al., 2013* and *Figure 1I*). *cima-1(wy84)* mutants result in defects in the ventral cephalic sheath cell (VCSC) glia position and morphology during growth (*Figure 1—figure supplement 1A–B*). Abnormal VCSC glia ectopically ensheath the normally asynaptic Zone 1 region of AIY, which causes ectopic presynaptic sites in Zone 1 that are not in apposition to AIY's wild-type postsynaptic partner, the RIA neurons (*Figure 1E–F,I*, *Figure 1—figure supplement 1C–F* and *Shao et al., 2013*). Therefore, in *cima-1* mutants, abnormal glia morphology and position during growth of the organism resulted in changes to the relationship between the glia and the neurite, which in turn disrupted the embryonically established synaptic pattern as the animal grew (*Figure 1F and I*). To identify molecules which cooperate with *cima-1* to regulate synaptic allometry, we performed an unbiased EMS screen in *cima-1(wy84)* mutants for suppressors of defects in the synaptic pattern, and isolated allele *ola226* (*Figure 2*).

Although the animal's morphology and the guidance of AIY neurites are largely unaffected in *cima-1(wy84);ola226* double mutants (*Figure 2—figure supplement 1A–C*), we found that *ola226* suppressed the ectopic distribution of both the vesicular marker RAB-3 and the active zone marker

SYD-1 in *cima-1(wy84)* (*Figure 1H*, and *Figure 2A–D'''*). Young *cima-1(wy84);ola226* animals displayed a wild-type pattern of presynaptic specializations (*Figure 1G*), suggesting that the *ola226* allele does not generally affect synaptogenesis. Instead, the *ola226* allele robustly suppresses the synaptic allometry defects observed in *cima-1(wy84)* mutants, as scored by the percentage of animals displaying ectopic presynaptic sites in the Zone 1 region and the relative presynaptic length in the neurite (93.9% of animals displayed ectopic presynaptic sites in *cima-1(wy84)* vs 54.6% in *cima-1(wy84);ola226* double mutants, p<0.0001; *Figure 2E–F*). Together, these results indicate that the *ola226* allele is specifically required for the suppression of the ectopic presynaptic specializations that form post-embryonically in the *cima-1(wy84)* mutants.

## Mutant allele *ola226* suppresses glia position and morphology defects in *cima-1* mutants

The emergence of ectopic presynaptic sites in *cima-1(wy84)* mutants requires ventral cephalic sheath cell (VCSC) glia extension during growth (*Shao et al., 2013*). Therefore growth, and the size of the animal, affect the expressivity of the allometry phenotypes in *cima-1(wy84)* mutants. For example, shorter *dpy* mutants suppress *cima-1(wy84)* synaptic allometry defects, while the longer *lon* mutants enhance *cima-1(wy84)* synaptic allometry defects (*Figure 2—figure supplement 1D–I'* and *Shao et al., 2013*). We examined the size of *ola226* and *cima-1(wy84);ola226* adult mutant animals and determined that it is indistinguishable from wild-type animals (*Figure 2—figure supplement 1C*), indicating that the effects of *ola226* in the *cima-1(wy84)* phenotype is through mechanisms distinct from those regulating the general size of the animal during development.

Next, we examined if *ola226* could alter VCSC glia morphology. We labeled VCSC glia with mCherry in wild type and the mutants, and quantified VCSC glia position and morphology (*Figure 3*). Consistent with and extending our previous observations, we observed that the VCSC glia in *cima-1(wy84)* mutants displayed defects in both position and morphology during growth. As *cima-1* mutant animals grew, VCSC glia were posteriorly displaced, resulting in longer VCSC glia anterior processes (mean length of the VCSC glia anterior process: 113.35 μm in wild type, 127.53 μm in *cima-1(wy84)* mutants, p<0.0001. *Figure 3B,C,F*). *cima-1* mutants glia endfeet also abnormally extended posteriorly (mean length of VCSC glia endfeet: 45.52 μm in wild type and 51.47 μm in *cima-1(wy84)* mutants, p<0.0001. *Figure 3B,C,G*). These two defects changed the positions of VCSC glia relative to the AIY neurite, resulting in ectopic presynaptic sites in *cima-1* mutant animals (*Figure 3B', C', H*). The AIY ectopic presynaptic sites in *cima-1* mutant animals are not in apposition to the normal postsynaptic RIA neurons (*Figure 1—figure supplement 1C–F*). Ablation of VCSC glia suppressed the ectopic presynaptic phenotype in *cima-1* mutants (*Shao et al., 2013*), indicating the importance of glia for the emergence of these ectopic presynaptic sites that disrupt the embryonically derived pattern of synaptic connectivity.

*cima-1(wy84);ola226* double mutants suppressed VCSC glia position and endfeet morphology phenotypes (length of glia anterior process: 127.53 μm in *cima-1(wy84)* and 120.68 μm in *cima-1(wy84);ola226*, p<0.0001; length of VCSC glia endfeet: 51.47 μm in *cima-1(wy84)* and 45.19 μm in *cima-17(wy84);ola226*, p<0.0001. *Figure 3D,F–G*). In these double mutants, the suppression caused a reduction in the abnormal region of contact seen in *cima-1(wy84)* mutants for the AIY neuron and VCSC glia (88.70% in *cima-1(wy84)* and 33.67% in *cima-1(wy84);ola226*, p<0.0001. *Figure 3D', H*). Consequently, ectopic presynaptic specializations that arise during growth in the AIY Zone 1 of *cima-1* mutants were suppressed, resulting in a synaptic pattern similar to that observed for wild type animals (*Figure 3B', D'*). Our findings suggest that *ola226* is a genetic lesion that suppresses *cima-1(wy84)* ectopic presynaptic sites by regulating glia position and morphology during allometric growth.

To better understand the phenotype of *ola226*, we outcrossed *cima-1(wy84)* and examined the resulting VCSC glia and AIY synaptic phenotypes for just the *ola226* mutants. We found that *ola226* mutant animals do not display defects in the position of VCSC glia (length of glia anterior process: 113.35 μm in wild type and 113.68 μm in *ola226*, p=0.72. *Figure 3E,F*). However, *ola226* mutants did display a modest but significant defect in VCSC glia morphology, with shorter posterior end-feet in *ola226* animals as compared to wild-type animals (length of glia end-feet: 45.52 μm in wild type, 39.79 μm in *ola226* p<0.0001. *Figure 3E,G*). *ola226* mutants also displayed a concomitant defect in the position of AIY, as both the neurite and the soma were anteriorly displaced compared to wild type animals (*Figure 3—figure supplement 1*). This anterior displacement of VCSC glia and AIY are

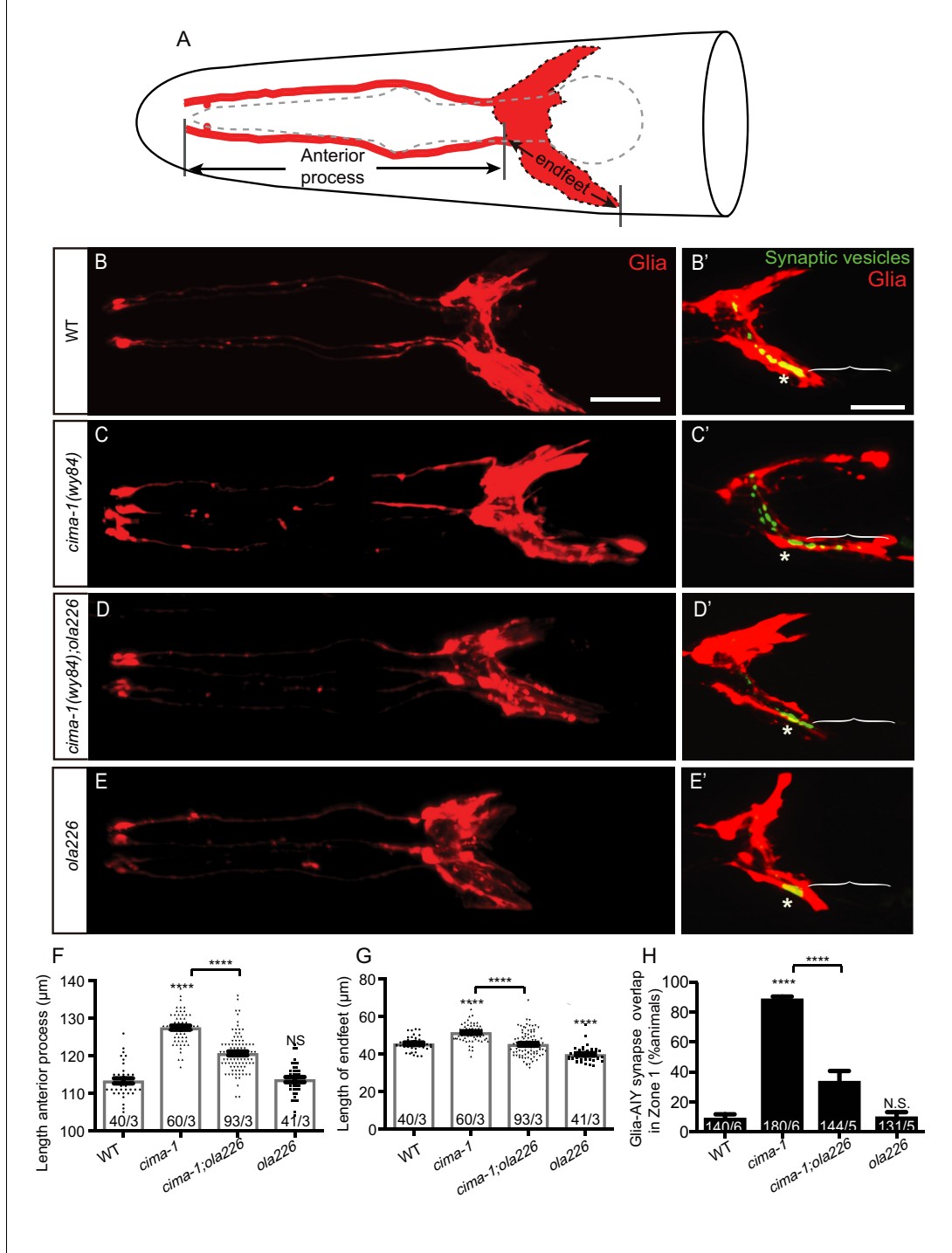

**Figure 3.** Glia morphology is affected in *ola226* mutants. (**A**) Cartoon diagram of the ventral and dorsal cephalic sheath cell glia (red) in the *C. elegans* head. The ventral cephalic sheath cell (VCSC) glia, located at the bottom half in the schematic, contacts the AIY synapses in the Zone 2 region. (**B–E'**) Confocal micrographs of the morphology of VCSC glia and the anterior process (red, labeled with *Phlh-17::mCherry*, (**B–E**), or VCSC glia cell body and endfeet (red) with the AIY presynaptic marker (green, GFP::RAB-3, (**B'–E'**) in adult wild type (**B, B'**), *cima-1(wy84)* mutants (**C, C'**), *cima-1(wy84);ola226* mutants (**D, D'**), and *ola226* mutants (**E, E'**). Brackets indicate the AIY Zone 1 region, and asterisks mark the AIY Zone 2 region (see **Figure 2A**). The animals imaged in B-E are not the same as B'-E'. (**F–H**) Quantification of phenotypes, including the length of glia anterior process (F, indicated in schematic A), the length of ventral endfeet (G, indicated in schematic A) and the percentage of animals displaying overlap between AIY synapses and VCSC glia in Zone 1 (H). The total number of animals (N) and the number of times scored (n) are indicated in each bar for each genotype as N/n. Statistical analyses are based on one-way ANOVA by Tukey's multiple comparison test. Error bars represent SEM, N.S.: not significant as compared to wild type, ****p<0.0001 as compared to wild type (if on top of bar graph), unless brackets are used between two compared genotypes.

*Figure 3 continued on next page*

*Figure 3 continued*

The online version of this article includes the following figure supplement(s) for figure 3:

**Figure supplement 1.** *ola226* affects AIY neurite and cell body position.

the opposite phenotype to that observed for *cima-1(wy84)* mutants, in which these cells are posteriorly displaced (*Shao et al., 2013*). Interestingly, unlike in *cima-1(wy84)* mutants, in the *ola226* mutants the area of overlap between the glia and AIY was not affected (*Figure 3E', H*). The distribution of presynaptic specializations in these animals was similar to that seen for wild type (*Figure 3B', E'*), consistent with the importance of glia position in sustaining presynaptic positions.

These phenotypes demonstrate that it is not just glia morphology, glia position or even the position of the AIY neurite in the animal that regulates synaptic allometry. Rather, the relative position between the VCSC glia and the AIY neurons appears to drive presynaptic positions during growth. Our data underscore the role of glia as guideposts in sustaining the synaptic pattern during post-embryonic growth.

## *ola226* is a lesion in *mig-17*, which encodes an ADAMTS metalloprotease

To identify which gene is affected in the *ola226* allele, we performed SNP mapping, whole genome sequencing and transgenic rescue experiments. The *ola226* allele results from a G to A mutation at the end of first exon of the *mig-17* gene and alters a conserved glutamic acid residue at position 19 to a lysine (*Figure 4A*). To test if *ola226* is a loss-of-function allele of *mig-17*, we examined two additional loss-of-function *mig-17* alleles, *mig-17(k113)* and *mig-17(k174)* (*Nishiwaki, 1999*; *Nishiwaki et al., 2000*). *mig-17(k113)* is a point mutation in the first intron of the gene and is predicted to affect correct splicing, while the *mig-17(k174)* allele results from a change in Q111 to a premature stop codon, producing a putative null allele (*Figure 4A*; *Shibata et al., 2016*). We found that just like *ola226*, both *k113* and *k174* alleles did not display phenotypes in the AIY presynaptic distribution on their own (*Figure 4—figure supplement 1*), yet robustly suppressed the ectopic presynaptic sites in *cima-1(wy84)* mutants (91.9% of animals displayed ectopic presynaptic sites in *cima-1(wy84)*, 62.3% in *cima-1(wy84);mig-17(k113)*, 29.9% in *cima-1(wy84);mig-17(k174)* and 45.7% in *cima-1(wy84);mig-17(ola226)*, p<0.0001 for all double mutants as compared to *cima-1(wy84)*; *Figure 4B–F,H*). Importantly, introducing a wild-type copy of the *mig-17* genomic sequence results in robust rescue of the *ola226* phenotype in *cima-1(wy84);mig-17(ola226)* double mutants (45.70% of animals displayed ectopic synapses in *cima-1(wy84);mig-17(ola226)* and 78.04% in *cima-1(wy84);mig-17(ola226);Pmig-17::mig-17*(genomic), p<0.0001; *Figure 4G,H*). Together our findings indicate that *ola226* is a recessive loss-of-function allele of *mig-17* which suppresses *cima-1(wy84)* defects in synaptic allometry by affecting glia positions during growth.

MIG-17 is an ADAMTS metalloprotease best known for its post-embryonic roles in regulating distal tip cell migration during gonad development (*Nishiwaki, 1999*) and pharyngeal size and shape during growth (*Shibata et al., 2016*). ADAMTS proteins have also been shown to regulate the basement membrane to maintain synaptic morphology at neuromuscular junctions (NMJs) (*Kurshan et al., 2014*; *Qin et al., 2014*). Careful examination of the pharynx length and the synaptic allometry defects in AIY revealed that the AIY synaptic allometry phenotypes do not simply arise from a defect in pharynx length (*Figure 4—figure supplement 2*). Unlike the NMJs, the basement membrane is not in direct contact with synapses in the nerve ring, including the AIY synapses (*White et al., 1986*). Therefore, the basement membrane cannot signal directly to AIY synapses as it does to the NMJs (*Kurshan et al., 2014*; *Qin et al., 2014*). Instead, our collective findings suggest that MIG-17 modulates synaptic allometry in AIY through the modulation of VCSC glia position and morphology.

## MIG-17 is expressed in muscles and neurons in the nerve ring

To examine how MIG-17 modulates synaptic allometry through the modulation of glia position and morphology, we next analyzed the expression pattern of *mig-17* in the nerve ring region. We found that a *mig-17* transcriptional GFP reporter was robustly expressed by body wall muscles as colabeled by P*myo-3::mCherry* (*Figure 5A–A'''* and consistent with *Nishiwaki et al., 2000*). We also observed

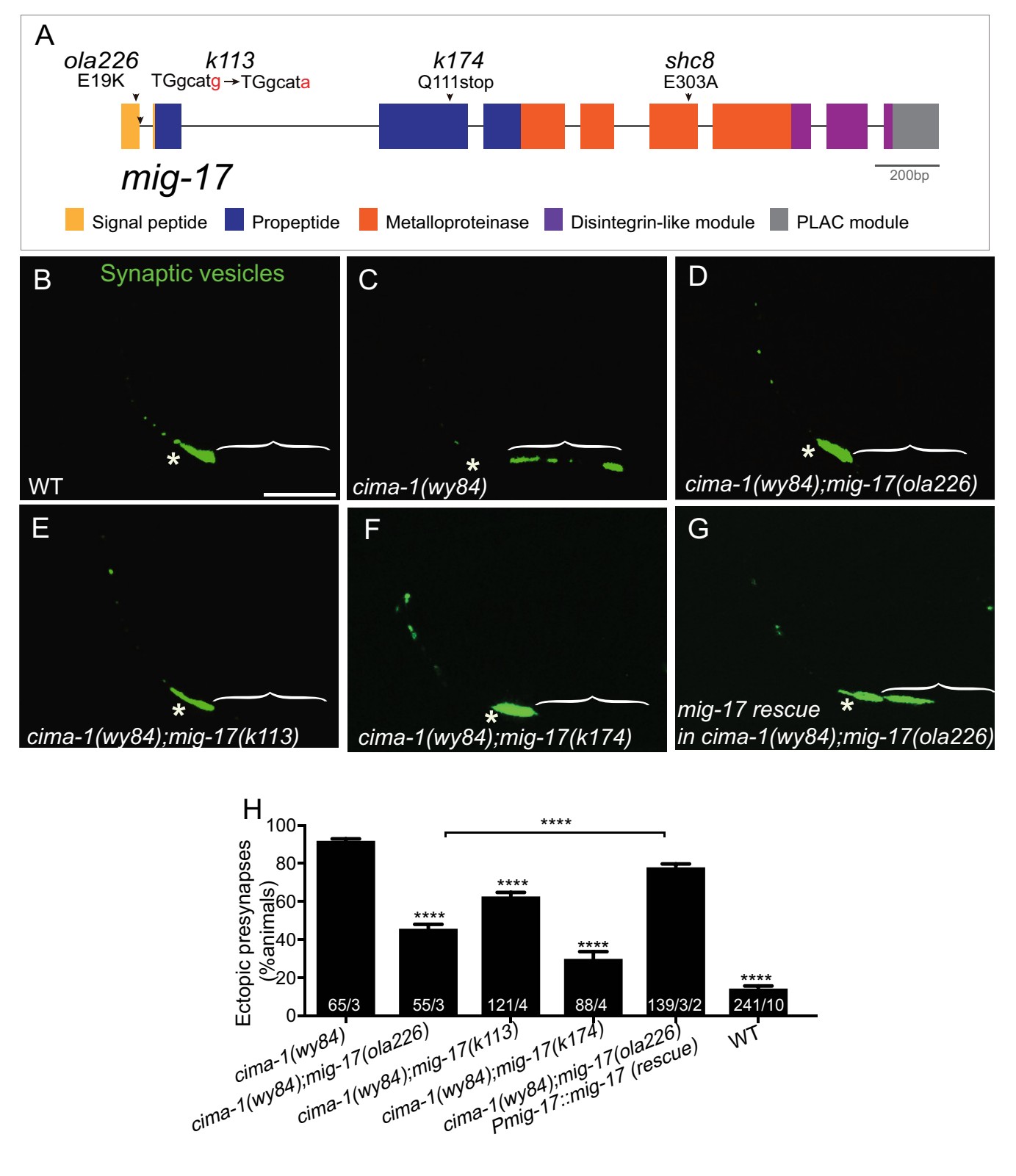

**Figure 4.** *ola226* is a lesion in the *mig-17* gene. (**A**) Schematic diagram of the *mig-17* gene and corresponding protein domains coded by the exons (colored) and genetic lesions for the alleles used in this study. (**B–G**) Confocal micrographs of the AIY synaptic vesicle marker GFP::RAB-3 (green) in adult wild type (**B**), *cima-1(wy84)* (**C**), *cima-1(wy84);mig-17(ola226)* (**D**), *cima-1(wy84);mig-17(k113)* (**E**), *cima-1(wy84);mig-17(k174)* (**F**), and *cima-1(wy84); mig-17(ola226)* animals expressing a wild-type copy of the *mig-17* gene (P*mig-17::mig-17(genomic)*) (**G**). Brackets indicate the AIY Zone 1 region.

*Figure 4 continued on next page*

*Figure 4 continued*

Asterisks indicate the Zone 2 region. Scale bar in (**B**) applies to all images, 10 µm. (**H**) Quantification of the percentage of animals with ectopic synapses in the AIY Zone 1 region for the indicated genotypes. The total number of animals (N) and the number of times scored (n1) are indicated in each bar for each genotype and for the transgenic lines created, the number of transgenic lines (n2) examined (all using the convention N/n1/n2). Statistical analyses are based on one-way ANOVA by Tukey's multiple comparison test. Error bars represent SEM, **p<0.01, ****p<0.0001 as compared to *cima-1 (wy84)* (if on top of bar graph), unless brackets are used between two compared genotypes.

The online version of this article includes the following figure supplement(s) for figure 4:

**Figure supplement 1.** Synaptic phenotypes in *mig-17* alleles.

**Figure supplement 2.** *mig-17(ola226)* and *cima-1(wy84)* phenotypes in pharyngeal length.

that in the head region, the reporter was detected in the nervous system (*Figure 5B–B'''*). We did not detect expression of MIG-17 in VCSC glial cells or in epidermal cells, where the MIG-17 genetic interactors CIMA-1 and EGL-15/FGFR are expressed (*Figure 5C–D'''*; *Shao et al., 2013*).

To determine the *mig-17* site of action, we expressed *mig-17* in the two tissues that showed *mig-17* expression: the nervous system (using the *rab-3* promoter *Nonet et al., 1997*); and the body wall muscles using the *myo-3* promoter *Miller et al. (1983)*; *Miller et al. (1986)*. We found robust rescue of the *cima-1(wy84);mig-17(ola226)* phenotype when *mig-17* was expressed either in body wall muscles or in the nervous system (*Figure 5E*), consistent with MIG-17 being a secreted ADAMTS protease. Indeed, expression of MIG-17 from a number of different cell-specific promoters, including glia and epidermal cells in which we did not detect expression, all resulted in rescue (*Figure 5—figure supplement 1*). Together, our findings suggest that secreted MIG-17 modulates glia morphology and synaptic allometry.

## MIG-17 requires its metalloprotease activity to promote the formation of ectopic presynaptic sites in *cima-1(wy84)* mutants

MIG-17 is an ADAMTS metalloprotease which remodels the basement membrane (*Nishiwaki et al., 2000*). To determine if MIG-17 acts through its canonical role of remodeling the basement membrane to regulate synaptic allometry, we first examined if its metalloprotease enzymatic activity was required for promoting the formation of ectopic synapses in *cima-1(wy84)* mutants. We engineered an E303A point mutation at the metalloprotease catalytic site (*Nishiwaki et al., 2000*) via CRISPR/cas-9 to generate the *mig-17(shc8)* allele (*Figure 6A*, CRISPR strategy outlined in *Figure 6—figure supplement 1A* is based on *Dickinson et al., 2013*; *Nishiwaki et al., 2000*). We observed that our engineered *mig-17(shc8)* allele behaved like other *mig-17* loss-of-function alleles and suppressed ectopic synapses in *cima-1(wy84)* mutant animals (91.91% of animals displayed ectopic synapses in *cima-1(wy84)* vs 57.49% in *cima-1(wy84);mig-17(shc8)*, p<0.0001, *Figure 6B–E,H*). Consistent with this result, we also found that a transgene with the E303A (*mig-17(E303A)*) lesion is incapable of rescuing the *mig-17*-induced suppression in *mig-17(ola226);cima-1(wy84)* mutants (*Figure 6F–H*). These findings indicate that MIG-17 metalloprotease enzymatic activity is required for promoting the formation of ectopic synapses in *cima-1(wy84)* mutants, and are consistent with a model whereby MIG-17 remodels the basement membrane to modulate synaptic allometry during growth.

## MIG-17 regulates basement membrane proteins to modulate synaptic allometry

To determine if MIG-17 remodels the basement membrane to modulate synaptic allometry, we examined the proteome through liquid chromatography–tandem mass spectrometry (LC-MS/MS) analyses in wild type and *mig-17(ola226)* mutant animals. Consistent with the known importance of MIG-17 in remodeling the basement membrane in other biological contexts (*Kim and Nishiwaki, 2015*), we observed significant and reproducible differences in the protein levels of basement membrane components for *mig-17(ola226)* mutants compared to wild type, including EMB-9/Collagen IV α1 chain, LET-2/Collagen IV α2 chain, OST-1/Sparc, UNC-52/Perlecan, NID-1/nidogen, EPI-1/laminin-α, LAM-1/laminin-β, and LAM-2/laminin-γ (*Figure 7A* and *Supplementary file 1*).

EMB-9/Collagen IV α1 is a core component of the basement membrane regulated by ADAMTS proteins (*Graham et al., 1997*; *Guo et al., 1991*; *Sibley et al., 1993*) and plays important roles in post-embryonic neuromuscular junction morphology (*Kurshan et al., 2014*; *Qin et al., 2014*). We

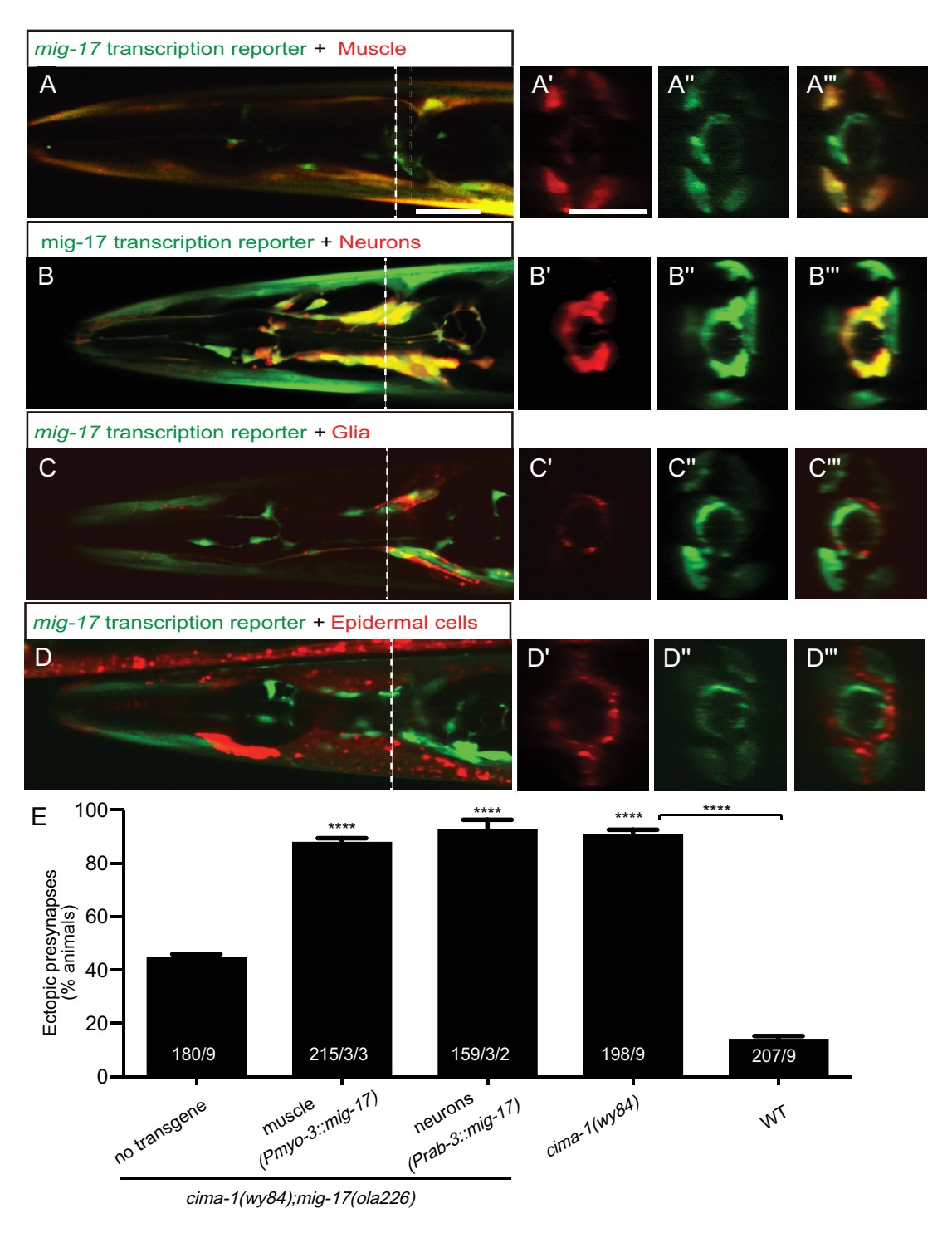

**Figure 5.** MIG-17 is a secreted molecule that regulates synaptic allometry. (A–D"') Confocal micrographs of adult animals expressing the transcriptional reporter *mig-17(genomic)::SL2::GFP* (green) with reporters that co-label body wall muscles (P*myo-3::mCherry* (A–A"')), neurons (P*rab-3::mCherry* (B–B"')), VCSC glia (P*hlh-17::mCherry* (C–C"')), epidermal cells (P*dpy-4::mCherry* (D–D"')). Images (A'–D"') correspond to a transverse cross-section of the confocal micrographs, specifically for the region corresponding to the dashed line in (A–D). The scale bar in (A) applies to (B, C, D), and in (A') applies all transverse cross-section images, and both scale bars are 10 μm. (E) Quantification of the percentage of adult animals with ectopic synapses in the AIY Zone 1 region of the indicated genotypes and rescue experiments. The total number of animals (N) and the number of times scored (n1) are indicated in each bar for each genotype, as are, for the transgenic lines created, the number of transgenic lines (n2) examined (all using the convention N/n1/n2). See also *Figure 5—figure supplement 1* for additional rescue experiments. Statistical analyses are based on one-way ANOVA by Tukey's

*Figure 5 continued on next page*

*Figure 5 continued*

multiple comparison test. Error bars represent SEM, N.S.: not significant, ****p<0.0001 compared to the no-transgene control (if on top of bar graph), unless brackets are used between two compared genotypes.

The online version of this article includes the following figure supplement(s) for figure 5:

**Figure supplement 1.** Cell-specific expression of *mig-17* in multiple tissues rescues the *mig-17* suppression in *mig-17(ola226);cima-1(wy84)* mutants.

wondered whether the AIY presynaptic sites, which have a different relationship to BM than do NMJs, would have altered morphology in *emb-9* mutant animals. Since EMB-9 null alleles are embryonic lethal (*Guo et al., 1991*; *Gupta et al., 1997*), we used neomorphic or hypomorphic missense alleles that disrupt NMJ morphology and are predicted to produce overabundant or disorganized collagen (*Gotenstein et al., 2018*; *Gupta et al., 1997*; *Kubota et al., 2012*; *Kurshan et al., 2014*; *Qin et al., 2014*). We did not observe detectable defects in the AIY presynaptic site distribution or morphology in *emb-9(xd51)* or *emb-9(b189)* mutants (*Figure 7—figure supplement 1*).

Interestingly, EMB-9/Collagen IV can also become overabundant or disorganized in ADAMTs mutants (*Kim and Nishiwaki, 2015*). We therefore hypothesized that the neomorphic or hypomorphic alleles of *emb-9* could phenocopy *mig-17* mutants and suppress the synaptic allometry defects for *cima-1* mutants. Indeed, we observed that neomorphic and hypomorphic *emb-9* alleles significantly suppressed the ectopic presynaptic sites in *cima-1(wy84)* mutant animals, although the penetrance of the suppression phenotype varied by the specific allele (*Figure 7B*). Therefore, while the *emb-9* alleles do not affect the morphology of AIY presynaptic sites (as they do for NMJ synapses), they significantly suppress the synaptic allometry defects for *cima-1* mutants.

We hypothesized that *cima-1* mutants are suppressed both by *mig-17* and the neomorphic and hypomorphic *emb-9* alleles because in these mutants the material properties of the basement membrane are altered. This, in turn, would prevent the movement of glia during growth and suppress the ectopic contacts between glia and AIY seen for *cima-1* mutants. If our hypothesis were correct, we would expect that other molecules known to modulate the levels or conformation of EMB-9 would also similarly affect synaptic allometry, as basement membrane properties would be altered. To test this, we imaged AIY presynaptic sites in alleles of *unc-52*/Perlecan and *fbl-1*/Fibulin, both of which can regulate the trafficking or function of EMB-9 (*Kubota et al., 2004*; *Kubota et al., 2012*; *Morrissey et al., 2016*; *Qin et al., 2014*). Consistent with our model, loss-of-function alleles of *unc-52*/Perlecan, which is known to functionally antagonize EMB-9/Collagen IV (*Qin et al., 2014*), significantly suppressed the ectopic presynaptic sites observed in *cima-1(wy84)* mutants (*Figure 7—figure supplement 1* and *Figure 7B*). Similarly, the gain-of-function *fbl-1(k201)* allele (*Kubota et al., 2004*), which is predicted to cause an overabundance of EMB-9 (*Kubota et al., 2012*), also suppressed the ectopic presynaptic sites observed in *cima-1(wy84)* mutants (*Figure 7—figure supplement 1* and *Figure 7B*). We also determined that the levels of suppression in *cima-1(wy84);mig-17(ola226);fbl-1(k201)* are similar to those seen in either *cima-1(wy84);mig-17(ola226)* or in *cima-1(wy84);fbl-1(k201)* mutants, consistent with *mig-17* and *fbl-1* genetically acting in the same pathway (*Figure 7B*).

Our findings indicate that while the different alleles of *emb-9* and *emb-9*-regulators might have different effects on the conformation or levels of the EMB-9 protein in the basement membrane, they all suppress the ectopic presynaptic site phenotype in *cima-1(wy84)* mutants. Their shared ability to suppress *cima-1(wy84)* mutants suggests that lesions resulting in defects in the basement membrane might prevent the repositioning of glia that gives rise to the ectopic presynaptic sites in *cima-1(wy84)* mutants.

## MIG-17 regulates EMB-9/Collagen IV α1 during post-embryonic growth

To better elucidate the relationship between MIG-17 and EMB-9 during growth, we examined MIG-17 and EMB-9 protein levels in vivo during post-embryonic development using an EMB-9::mCherry translational reporter (*Ihara et al., 2011*) and a MIG-17::mNeonGreen knock-in allele (via CRISPR-Cas9 strategies as described in *Dickinson et al., 2013*; *Figure 6—figure supplement 1B*). We observed that both MIG-17 and EMB-9 localize in the head-region to a pattern reminiscent of the extracellular matrix proximal to the pharynx bulb (*Ihara et al., 2011*). We also determined that the levels of MIG-17 and EMB-9 were regulated during post-embryonic growth. MIG-17 protein levels were detectable in larva stage one through larva stage 4, but became undetectable upon reaching

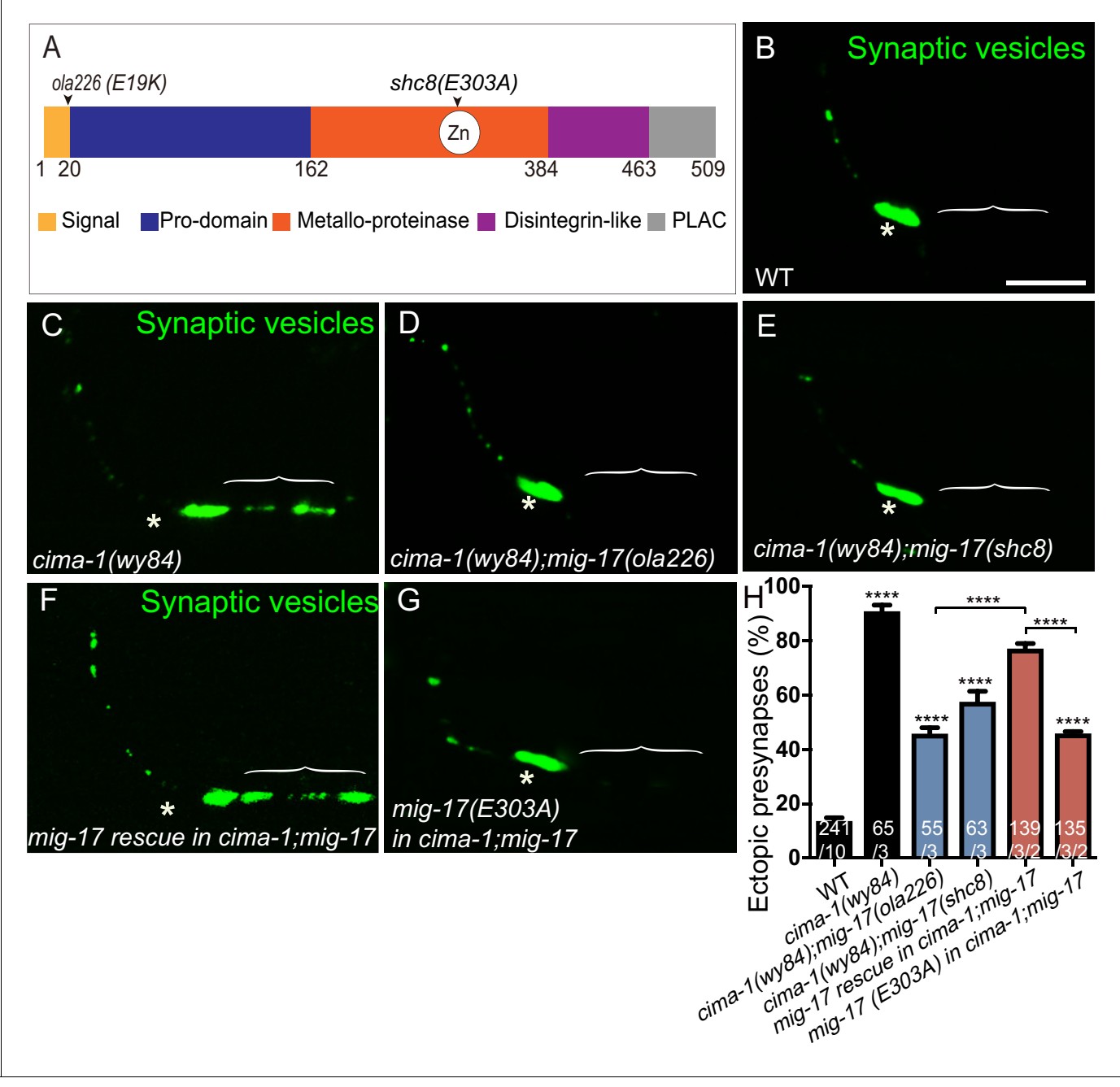

**Figure 6.** The metalloprotease activity of MIG-17 is required to suppress the formation of ectopic synapses in *cima-1(wy84)* mutants. (A) Schematic diagram of the MIG-17 protein, corresponding conserved protein domains (colored) and genetic lesions for the alleles used in this study. (B–G) Confocal micrographs of the AIY presynaptic sites labeled with the synaptic vesicle marker GFP::RAB-3 (pseudo-colored green) in adult wild type (B), *cima-1(wy84)* (C), *cima-1(wy84);mig-17(ola226)* (D), *cima-1(wy84);mig-17(shc8)* (E), *cima-1(wy84);mig-17(ola226)* animals expressing a wild type copy of the *mig-17* genomic DNA (P*mig-17::mig-17*) (F), and *cima-1(wy84);mig-17(ola226)* animals expressing a copy of the *mig-17* genomic DNA with a point mutation in the metalloprotease domain (P*mig-17::mig-17(E303A)*) (G). Brackets indicate the AIY Zone 1 region, and asterisks indicate the Zone 2 region. The scale bar in (B) is 10 µm and applies to all images. (H) Quantification of the percentage of animals with ectopic synapses in the AIY Zone 1 region in the indicated genotypes. In the graph, the transgene rescue with wild type copy of the *mig-17* genomic DNA control data is the same as in *Figure 4H*. The total number of animals (N) and the number of times scored (n1) are indicated in each bar for each genotype, as are, for the transgenic lines created, the number of transgenic lines (n2) examined (all using the convention N/n1/n2). Bars are pseudocolored by experiments, with controls in black, comparisons across *mig-17* alleles in blue and rescue experiments in red. Statistical analyses are based on one-way ANOVA by Tukey's multiple comparison test. Error bars represent SEM, N.S.: not significant, ****p<0.0001 compared to wild type (if on top of bar graph), unless brackets are used between two compared genotypes.

*Figure 6 continued on next page*

*Figure 6 continued*

The online version of this article includes the following figure supplement(s) for figure 6:

**Figure supplement 1.** CRISPR strategies to generate the *mig-17(shc8)* allele and the endogenous MIG-17::mNeonGreen.

the adult stage (*Figure 7C–G*, these results are consistent with previous in situ and western blot studies; *Ihara and Nishiwaki, 2008*). Conversely, EMB-9 protein levels increased as animals progress through the larval stages, achieving maximal expression in the adult stage (*Figure 7H–L*). Therefore, during post-embryonic growth, high protein levels of MIG-17 correlate with low protein levels of EMB-9.

The in vivo characterization of the protein levels of MIG-17 and EMB-9 are consistent with our proteomic results, and suggest that, directly or indirectly, MIG-17 regulates EMB-9 and basement membrane properties. Consistent with these findings, EMB-9::mCherry levels in *mig-17(ola226)* mutant animals were upregulated as compared to wild type (*Figure 7M–Q*). Interestingly, this increase in EMB-9 levels observed for *mig-17(ola226)* mutant was suppressed in *cima-1(wy84);mig-17(ola226)* double mutants, suggesting the existence of other *cima-1*-dependent mechanisms that modulate EMB-9 levels in the absence of MIG-17 (*Figure 7P and Q*). Importantly, our observations indicate that MIG-17 regulates EMB-9 and basement membrane properties to modulate synaptic allometry during post-embryonic growth.

Together, our findings support a model in which secreted metalloprotease MIG-17, whose levels are regulated during post-embryonic growth, dynamically regulates the muscle-derived basement membrane. Through regulation of the basement membrane, MIG-17 modulates *cima-1*-dependent epidermal-glial crosstalk to regulate glia position and morphology and sustain synaptic allometry during growth.

## MIG-17 and EGL-15/FGFR promote ectopic presynaptic site formation in *cima-1(wy84)*

CIMA-1 modulates epidermal-glial cell adhesion via regulation of EGL-15/FGFR ectodomain which acts, not in its canonical signaling role, but as an extracellular adhesion factor (*Bülow et al., 2004*; *Shao et al., 2013*). Consistent with this model, CIMA-1 is required to regulate EGL-15(5A)/FGFR protein levels, and overexpression of the EGL-15(5A)/FGFR ectodomain in wild-type animals phenocopied *cima-1* mutants (*Shao et al., 2013*). What is the relationship between MIG-17 and the glia-epidermis contacts modulated by CIMA-1 and EGL-15(5A)/FGFR?

We first examined if *mig-17* mutants could enhance *egl-15/*FGFR suppression of *cima-1*. We determined that *cima-1(wy84);egl-15(n484)* double mutants, *cima-1(wy84);mig-17(ola226)* double mutants and *cima-1(wy84);mig-17(ola226);egl-15(n484)* triple mutants all suppressed the *cima-1 (wy84)* phenotype of ectopic presynaptic sites in a similar manner (*Figure 8A–D*). We note that while the observed suppression was not a complete reversion to wild-type phenotypes, it is consistent with the degree of suppression observed for glia-ablated animals (*Shao et al., 2013*). Importantly, these findings indicate that alleles for *mig-17* and *egl-15* similarly suppress the *cima-1* phenotype, and are incapable of enhancing each other's effect on the suppression of *cima-1*, consistent with them acting in different tissues, but in similar genetic pathways to suppress *cima-1* mutant defects in synaptic allometry.

To further probe the relationship between EGL-15(5A)/FGFR and MIG-17, we examined synapses and glia in animals overexpressing EGL-15(5A)/FGFR. Overexpression of EGL-15(5A)/FGFR in epidermal cells promotes VCSC glia end-feet extension and ectopic presynaptic sites in AIY. This result phenocopies *cima-1(wy84)* mutants, and supports the idea that *cima-1* acts antagonistically to the EGL-15/FGF Receptor (*Figure 8F,H,I* and *Shao et al., 2013*). Interestingly, we observed that *mig-17 (ola226)* suppresses VCSC glia extension and the AIY ectopic presynaptic sites that arise during post-embryonic growth in animals over-expressing EGL-15(5A)/FGFR (*Figure 8G–I*). This result is consistent with MIG-17 and EGL-15/FGFR acting in the same inter-tissue synaptic allometry pathway.

Together, our genetic findings indicate that EGL-15(5A)/FGFR and MIG-17 genetically interact to position glia and regulate synaptic allometry during growth (*Figure 8J*). The finding that *mig-17 (ola226)* suppresses VCSC glia extension and ectopic synapses in animals over-expressing EGL-15 (5A)/FGFR indicates that *mig-17* is epistatic to *egl-15*. While we cannot exclude the possibility that

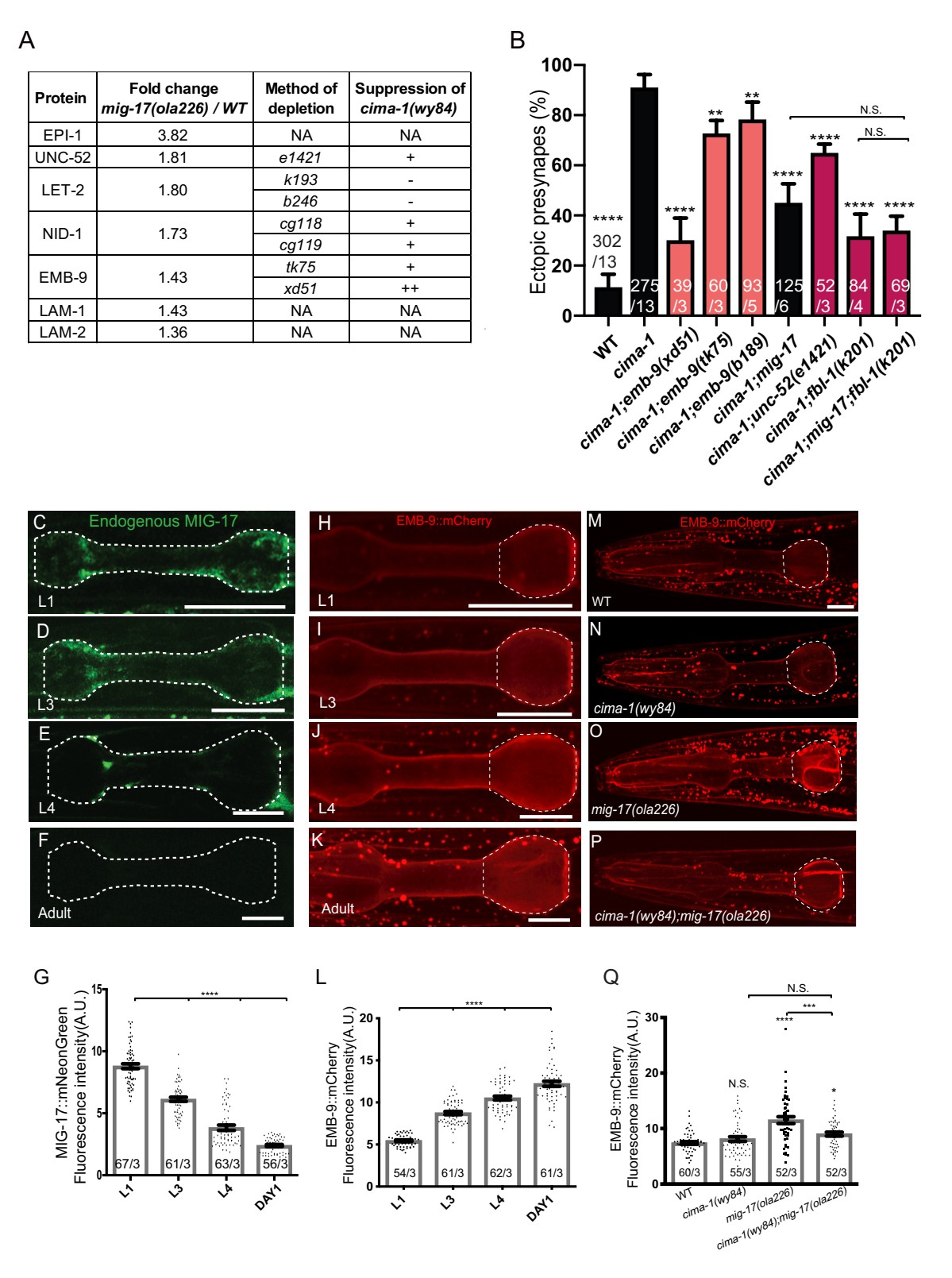

**Figure 7.** MIG-17 modulates synaptic allometry through the regulation of the basement membrane. (**A**) List of basement membrane components upregulated in the mass spectrometry analyses (see also **Supplementary file 1**), and alleles tested with *cima-1* for their capacity to suppress the synaptic allometry phenotypes in adult worms. (**B**) Quantification of the percentage of animals with ectopic synapses in the Zone 1 region of AIY for the indicated the genotypes. Bars are pseudocolored by experiment, with black bars corresponding to controls, light pink bars corresponding to *emb-9*

*Figure 7 continued on next page*

*Figure 7 continued*

alleles, and red bars corresponding to alleles for genes known to regulate *emb-9*, such as *unc-52* and *fbl-1*. (C–F) Confocal micrographs of the pharynx (dashed line) of animals with a CRISPR-engineered MIG-17::mNeonGreen imaged at larva stage 1 (C), larva stage 3 (D), larva stage 4 (E) and 1 day-old adults (F) in wild-type animals. (G) Quantification of the average MIG-17::mNeonGreen intensity in the pharyngeal area (outlined with dashed lines in C-F) at the indicated developmental stages. (H–L) As (C–G), but imaging an integrated EMB-9::mCherry strain (a gift from David Sherwood) in wild type animals. (M–Q) As (H–K) but in adults of wild type (M); *cima-1(wy84)* (N); *mig-17(ola226)* (O); *cima-1(wy84);mig-17(ola226)* (P) and quantified in (Q). The statistics are based on one-way ANOVA by Tukey's multiple comparison test. In the graphs, the total number of animals (N) and the number of times scored (n) are indicated in each bar for each genotype as N/n. Error bars represent SEM, N.S.: not significant, **$p<0.01$, ***$p<0.001$, ****$p<0.0001$ for indicated comparison. For all images, scale bars are 10 µm. The scale bar in (M) applies to (N–P).

The online version of this article includes the following figure supplement(s) for figure 7:

**Figure supplement 1.** Phenotypes in AIY synapses of alleles affecting basement membrane proteins.

EGL-15(5A)/FGFR is a substrate of MIG-17, their epistatic relationship suggests that *mig-17* acts downstream (or in parallel) to modulate the role of *egl-15* in positioning glia and regulating synaptic allometry (*Figure 8J*). Together with our other findings, we favor a model whereby MIG-17 modifies the basement membrane to modulate the effects of CIMA-1 and EGL-15 regulated epidermal-glial crosstalk on glia location and morphology during growth.

## VCSC Glia bridge epidermal-derived growth signals with the muscle-secreted basement membrane to sustain synaptic allometry

How do these molecules, which are derived from non-neuronal tissues (muscle cells and epidermal cells) that do not contact the synapses act together to regulate synaptic allometry? To understand this, we examined electron micrographs and fluorescent microscopy images that show the anatomical relationship among synapses in AIY interneurons, VCSC glia, epidermal cells, basement membrane and muscles (*Altun, 2019*; *White et al., 1986*).

The AIY Zone 2 synaptic region lies in the ventral base of the nerve ring bundle and is in direct contact with the nerve ring-facing side of VCSC glia (*Altun, 2019*; *White et al., 1986*). No basement membrane is observed between VCSC glia and nerve ring neurons (*Figure 9* and *Figure 9—figure supplement 1*). On the pseudocoelom-facing side, VCSC glia contact two distinct non-neuronal tissues: epidermal cells and muscle-derived basement membrane. VCSC are in direct contact with epidermal cells, which regulate glia morphology during growth through the expression of CIMA-1 and the EGL-15/FGF Receptor (*Figure 9A*, *Figure 9—figure supplement 1D* and *Shao et al., 2013*). No basement membrane is observable between VCSC glia and epidermal cells (*Figure 9B–D*, *Figure 9—figure supplement 1D*). However, we observed that at regions where glia are apposed to muscle cells, VCSC glia were decorated with basement membrane on the side facing the pseudocoelom cavity (*Figure 9B–D*, *Figure 9—figure supplement 1D*). Thus, VCSC glia have three surface regions: direct contact with neurons (on the nerve ring-facing side), direct contact with the epidermal cells (on the pseudocoelom-facing side), and contact with muscle-derived basement membrane (also on the pseudocoelom-facing side) (*Figure 9D*, *Figure 9—figure supplement 1D*).

Our data collectively indicate that secreted MIG-17 modulates the basement membrane. Regulation of the basement membrane by MIG-17 during post-embryonic growth acts in opposition to the CIMA-1-mediated epidermal-glial crosstalk. Therefore, muscles and epidermal cells interact with glia on the pseudocoelom-facing side and cooperate to regulate glia position (and morphology) during growth. Glia contact synapses on their nerve ring-facing side and sustain synaptic positions. Our data suggest that glia act as guideposts during growth, translating growth information from epidermal cells and muscles to guide synaptic allometry and preserve the embryonically-derived synaptic patterns during post-embryonic growth.

## Discussion

We uncovered a muscle-epidermis-glia signaling axis, modulated by *mig-17* and the basement membrane, which sustains synaptic allometry during growth. Suppressor forward genetic screens in the *cima-1* mutant background identified *mig-17*, which encodes a secreted ADAMTS metalloprotease (*Nishiwaki et al., 2000*). We found that secreted *mig-17* modulates basement membrane proteins. The basement membrane does not directly contact the affected synapses. Instead, muscle-derived

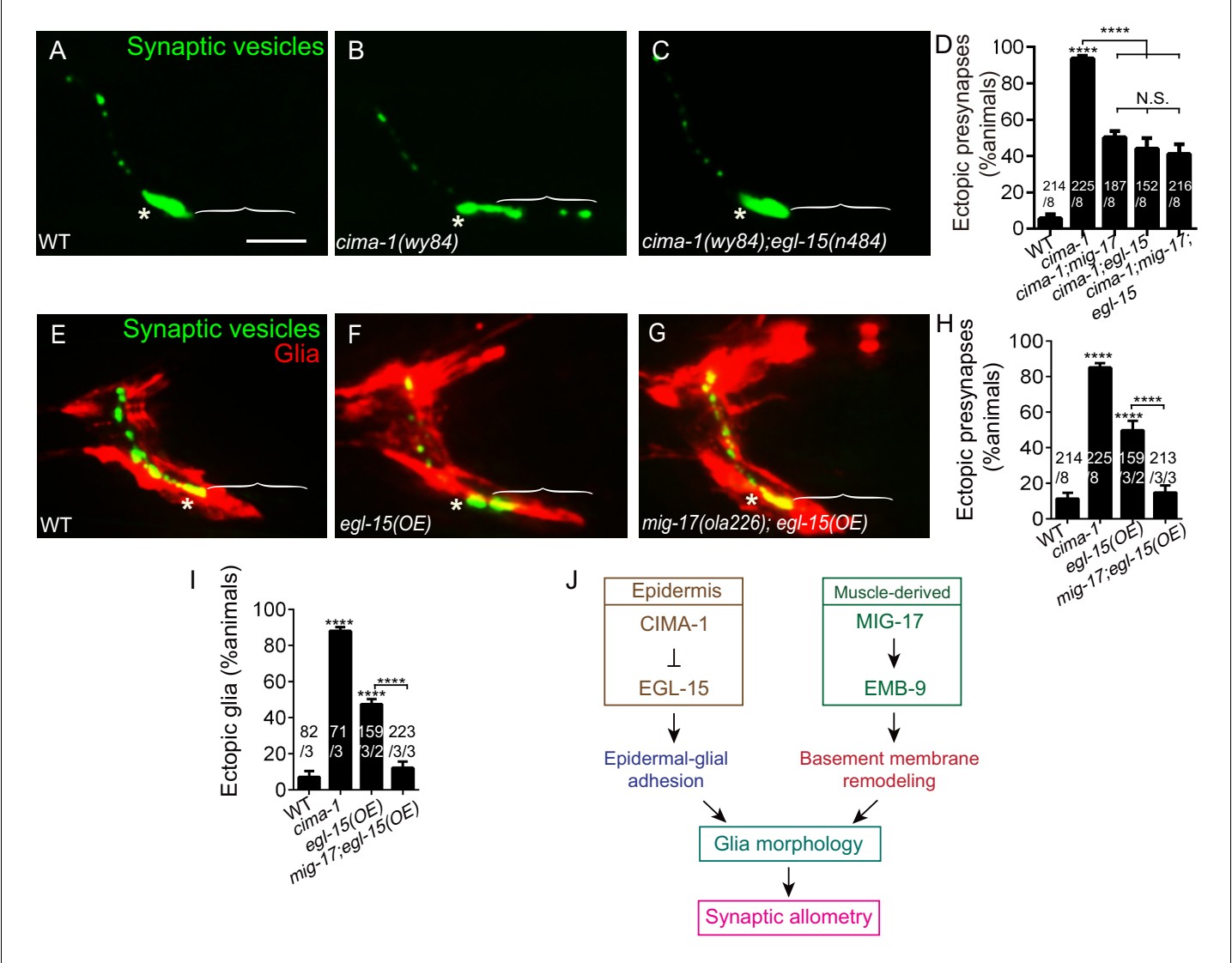

**Figure 8.** MIG-17 genetically interacts with EGL-15/Fibroblast Growth Factor Receptor to regulate synaptic allometry. (A–C) Confocal micrographs of the AIY synaptic vesicle marker GFP::RAB-3 (green) in adult wild type (A), *cima-1(wy84)* (B), *cima-1(wy84);egl-15(n484)* (C). (D) Quantification of percentage of animals with ectopic synapses in the indicated genotypes. (E–G) Confocal micrographs of AIY synaptic vesicle marker GFP::RAB-3 (green) and VCSC glia (red) in adult wild type (E), wild-type animals overexpressing EGL-15(isoform 5A) in epidermal cells by using P*dpy-7::egl-15(5A)* (F) and *mig-17(ola226)* overexpressing EGL-15(isoform 5A) in epidermal cells by using P*dpy-7::egl-15(5A)* (G). (H–I) Quantification of percentage of animals with ectopic synapses (H) or ectopic glia (I) in the indicated genotypes. (J) Schematic model of the multi-tissue regulation of synaptic allometry in AIY, as in *Figure 1I*, but with the new findings on *mig-17*. In all images (A–C, E–G), brackets indicate the AIY Zone 1 region, asterisks mark the Zone 2 region. Scale bar in (A), 10 μm, applies to all images. In the graphs (D, H, I), the total number of animals (N), the number of times scored (n1) are indicated in each bar for each genotype, as are, for the transgenic lines created, the number of transgenic lines (n2) examined (all using the convention N/n1/n2). Statistical analyses are based on one-way ANOVA by Tukey's multiple comparison test. Error bars represent SEM, N.S.: not significant, **p<0.01, ***p<0.001, ****p<0.0001 as compared to wild type (if on top of bar graph), unless brackets are used between two compared genotypes.

basement membrane coats the pseudocoelum-facing side of glia, while glia contact synapses on their other cellular side. MIG-17 is regulated during growth and remodels the basement membrane to modulate glia morphology, which then modulates presynaptic positions during growth. Our findings underscore the critical role of non-neuronal cells in sustaining synaptic allometry in vivo.

Glia act as guideposts to regulate presynaptic positions during growth. We previously demonstrated that glia play critical roles, both during embryonic development and during post-embryonic growth, to sustain presynaptic positions in *C. elegans*. During embryonic development, VCSC glia

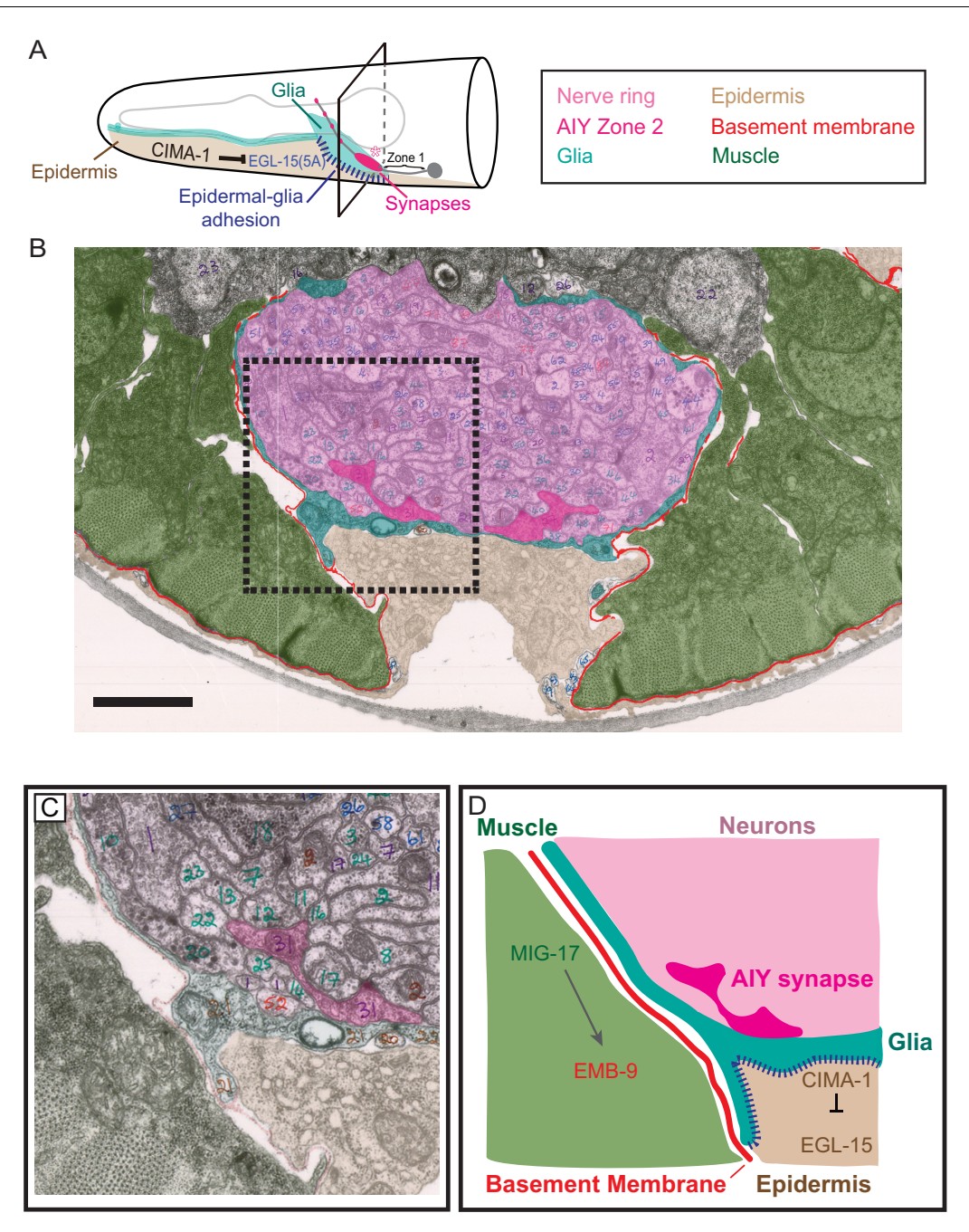

**Figure 9.** Glia maintain synaptic allometry by bridging epidermal-derived growth signals with the muscle-secreted basement membrane. (A) Schematic of the head of *C. elegans*, as in *Figure 1C*, with indicated tissues pseudocolored. Box corresponds to cross sections examined in (B–D). (B) Segmented electron micrograph from a wild type animal (JSH236 from *White et al., 1986*). The EM corresponds to the Zone 2 region of AIY with muscles (pseudo-colored green), basement membrane (BM, pseudo-colored red), VCSC glia (pseudo-colored teal), epidermal cell (pseudo-colored beige) and the ventral bundle of the nerve ring (pseudo-colored pink, including AIY Zone two pseudo-colored dark pink). (C) Zoom-in of the dashed-boxed region in (B). The pseudo-coloring opacity is decreased as to show that the basement membrane is specifically observed between muscle and VCSC glia, but not between glia and epidermal cells or between glia and neurons. (D) A cartoon diagram depicting the cross-section of the *C. elegans* nerve ring as shown in (C) (modified from WormAtlas.org), and represented as a molecular and cellular model of our in vivo data regarding the role of non-neuronal cells in glia position and morphology to regulate synaptic allometry during growth. As illustrated in the cartoon and the EM image, body wall muscle (green), the nerve ring (pink) and glia (teal) are proximal to the epidermal cells (beige). The nerve ring bundle is surrounded by VCSC glia, which contact it directly. At the other side of the glia cell, it faces the pseudocoelum and interacts with muscle-derived basement membrane (red) and epidermal cells (beige).

*Figure 9 continued on next page*

*Figure 9 continued*

The online version of this article includes the following figure supplement(s) for figure 9:

**Figure supplement 1.** Localization of basement membrane near the nerve ring.

secrete a chemotrophic factor (Netrin) to coordinate synaptic spatial specificity between AIY and its post-synaptic partner, called RIA (*Colón-Ramos et al., 2007*). Notably, postsynaptic RIA is not necessary for AIY to correctly establish the position of presynaptic specializations, underscoring the role of non-neuronal cells in presynaptic positioning, and coordinated synapse assembly, during development (*Colón-Ramos et al., 2007*). During post-embryonic growth, the same VCSC glia are required to sustain presynaptic positions but through distinct, Netrin-independent signaling pathways (*Shao et al., 2013*). Our current study demonstrates that sustaining synaptic allometry depends on the relative position of the glia end-feet with respect to the AIY neurite. By using genetic and in vivo cell biological manipulations, we could alter the position of both VCSC glia and AIY. Even when both cells were mispositioned in the animal, as long as their contact relationship was sustained, correct synaptic allometry was sustained (*Figure 3E–H*). Our findings are consistent with vertebrate and invertebrate studies supporting essential roles for glia in regulating synaptic assembly and function in vivo (*Allen and Eroglu, 2017*; *Van Horn and Ruthazer, 2019*). We extend these findings to highlight a role for glia in sustaining the embryonically established synaptic pattern during post-embryonic allometric growth.

Glia morphology and positions are actively maintained during growth. Growth in *C. elegans* relies on coordinated signals from epidermal cells and body wall muscles (*Chisholm and Hardin, 2005*). Epidermal cells express genes that regulate molting, body morphogenesis and animal size (*Chisholm and Hsiao, 2012a*; *Chisholm and Xu, 2012b*). Body wall muscle contractions regulate elongation during embryogenesis, and influence epidermal cytoskeletal remodeling via tension-sensing mechanisms (*Chisholm and Hsiao, 2012a*; *Chisholm and Xu, 2012b*; *Williams and Waterston, 1994*). While we do not yet understand how organisms sense growth, our findings uncovered a cooperative signaling pathway that emerges from these two growth-regulating cell types to position glia, which then drives synaptic positioning during allometry. Our genetic studies demonstrate that secreted MIG-17 is epistatic to epidermally derived CIMA-1 and EGL-15/FGFR. These results show a multi-tissue, non-neuronal pathway that converges to transduce growth information and position glia to regulate synaptic allometry. Thus, our findings uncover a non-cell autonomous, two-component system that cooperates to transduce growth information to the nervous system through glia.

During post-embryonic growth, ADAMTS protease MIG-17 regulates the basement membrane to modulate synaptic allometry. In *Drosophila*, the development of the peripheral nervous system and the maintenance of central nervous system architecture require homologous ADAMTS Stl and AdamT-A proteins (*Lhamo and Ismat, 2015*; *Skeath et al., 2017*). In general, ADAMTS metalloproteases function to degrade and remodel the extracellular matrix (*Krishnaswamy et al., 2019*). In humans, lesions in ADAMTS genes produce biomedically important defects, including short stature and neuronal developmental disorders, among other problems (*Cheng et al., 2018*; *Howell et al., 2012*; *Miguel et al., 2005*). Remodeling the extracellular matrix in *C. elegans* also contributes to gonad organogenesis and pharynx growth. These processes are partially mediated by the MIG-17 metalloprotease (*Kim and Nishiwaki, 2015*; *Kubota et al., 2004*; *Kubota et al., 2008*; *Nishiwaki et al., 2000*; *Shibata et al., 2016*).

Our proteomic, genetic and cell biological findings strongly suggest that the basement membrane is a dynamic structure that remodels, and that MIG-17 regulates synaptic allometry by modulating the basement membrane. Common among the genetic manipulations presented here—loss-of-function *mig-17* and *unc-52* alleles, gain of function *fbl-1* alleles or hypomorphic and neomorphic *emb-9* alleles—is a resulting disorganized basement membrane. All these alleles also suppress the ectopic synapses observed for *cima-1* mutants. We hypothesize that these alleles all suppress *cima-1* mutants because the material properties of the basement membrane prevent the movement of the glia during growth. This inability to reposition does not disrupt synaptic allometry as long as the glia and AIY relationship is preserved, as is the case in *mig-17* and other basement membrane single mutants. But synaptic allometry defects occur when the relationship between glia and the AIY neurite is altered, as in the *cima-1* mutants, in which epidermal-glia adhesion abnormally extends glia

posteriorly. Therefore, MIG-17 and the basement membrane proteins act in opposition to CIMA-1 in positioning glia and regulating synaptic allometry during growth.

Our results demonstrate that modulating glia morphology and synaptic positions requires a muscle-epidermis-glia signaling axis, which utilizes MIG-17 dependent regulation of the extracellular matrix. We note that while basement membrane proteins can also regulate neuromuscular junction synapses (*Ackley et al., 2003*; *Kurshan et al., 2014*; *Patton, 2003*; *Qin et al., 2014*; *Rogers and Nishimune, 2017*), NMJs are in direct contact with the basement membrane. The neurons examined in this study, which are in the nerve ring, are not in direct contact with the basement membrane (*White et al., 1986*). Instead VCSC glia ensheath the nerve ring to form a physical barrier between the neuropil and adjacent tissues, including the pseudocoelom, the basement membrane and the epidermal cells (*Shaham, 2015*). At one side, VCSC glia contact neurons in the nerve ring, while at the other side they are either decorated by basement membrane or in direct contact with epidermal cells. Interactions among the VCSC glia, basement membrane and epidermal cells reflect the genetic relationships we uncovered in our forward genetic screens, as epidermal CIMA-1 and EGL-15/FGFR modulate glia morphology through epidermal-glial adhesion, and secreted MIG-17 modulate glia morphology through the muscle-derived extracellular matrix.

The muscle-epidermis-glia signaling axis described here is reminiscent of the neurovascular unit of the blood-brain barrier in *Drosophila* and vertebrates. In the vertebrate neurovascular unit, muscle-related pericyte cells interact with vascular endothelial cells and astrocytes through the basement membrane (*Xu et al., 2019*). Pericytes, endothelial cells and the basement membrane are not in direct contact with neurons. Instead, astrocytes mediate signaling between these non-neuronal cells and neurons, including coupling the developmental programs that coordinate vasculature development and neurodevelopment (*Tam and Watts, 2010*), and the functional programs that coordinate neuronal activity with blood flow (*Allan, 2006*; *Koehler et al., 2009*). We note that the extracellular matrix of the blood-brain barrier is molecularly similar to the basement membrane of *C. elegans*, and includes molecules we tested here, such as laminin, collagen IV and fibulin (*Thomsen et al., 2017*). While the role of these components in vertebrate synaptic allometry has not been examined, we speculate that the functional neurovascular unit may transduce information from the vasculature to sustain synaptic positions during allometric growth. Our findings therefore uncover a novel muscle-epidermis-glia signaling axis, which communicates in part through the remodeling of the basement membrane to sustains synaptic specificity during the organism's allometric growth. We hypothesize that analogous structures in other organisms may represent conserved signaling axis that couple glia-mediated communication among non-neuronal cells and neurons to position synapses.

## Materials and methods

### Strains

All strains were grown at 22°C on NGM agar plates seeded with *Escherichia coli* OP50 (*Brenner, 1974*), except temperature sensitive strain *emb-9(b189)*, grown at 16°C until L1 stage and then transferred to 22.5°C (*Gupta et al., 1997*). *C. elegans* N2 bristol was used as the wild-type strain.

The following alleles were utilized in this study:

- LGII: *unc-52(gk3)*, *unc-52(e1421)*
- LGIII: *emb-9(tk75)*, *emb-9(xd51)*, *emb-9(b189))*
- LGIV: *cima-1(wy84)*, *fbl-1(k201)*, *dpy-4(e1166)*
- LGV: *mig-17(ola226)*, *mig-17(k113)*, *mig-17(k174)*, *mig-17(shc8)*, *mig-17(shc19)*, *nid-1(cg118)*, *nid-1(cg119)*, *lon-3(e2175)*
- LGX: *let-2(k193)*, *let-2(b246)*, *egl-15(n484)*

The following transgenic lines were used in this study: *shcEx1126*, *shcEx1127* and *shcEx1128[Pttx-3::syd-1::GFP;Pttx-3::rab-3::mCherry;Punc-122::RFP]*, *shcEx1146* and *shcEx1147[Pmig-17::mig-17 genomics;Phlh-17::mCherry]*, *shcEx1129[Pmig-17::mig-17::SL2::GFP;Pdpy-4::mCherry]*, *shcEx1130 [Pmig-17::mig-17::SL2::GFP;Pmyo-3::mCherry]*, *shcEx1131[Pmig-17::mig-17::SL2::GFP;Phlh-17:: mCherry]*, *shcEx1410[Pmig-17::mig-17::SL2::GFP;Prab-3::mCherry]*, *shcEx845[Phlh-17::mCherry]*, *shcEx1145[Pdpy-4::mCherry]*, *shcEx1402[Pmyo-3::mCherry]*, *shcEx1403[Prab-3::mCherry]*, *shcEx1414 and shcEx1415 [Pmig-17::mig-17(E303A); Phlh-17::mCherry]*, *shcEx1133*, *shcEx1134* and *shcEx1135*

[Pmyo-3::mig-17;Phlh-17::mCherry], shcEx1676, shcEx1677 and shcEx1678[Plim-4::mig-17;Phlh-17:: mCherry], shcEx1139 and shcEx1140[Phlh-17::mig-17;Phlh-17::mCherry], shcEx1142 and shcEx1143 [Pdpy-7::mig-17;Punc-122::GFP], shcEx1675, shcEx1684 and shcEx1685 [Pttx-3::mig-17;Phlh-17:: mCherry], qyIs46[unc119;emb-9::mCherry], shcEx776, shcEx777, shcEx778, shcEx780 and shcEx781 [Phlh-17::mCherry;Pttx-3::GFP::rab-3], shcEx424, shcEx425, shcEx536, shcEx537 and shcEx538[Pdpy-7::egl-15(5A);Phlh-17::mCherry;Pttx-3::GFP:: rab-3], shcEx1252 and shcEx1253 [Pmig-17::mig-17 (genomic);Phlh-17::mCherry], shcEx1682 and shcEx1683 [Prab-3::mig-17; Phlh-17::mCherry], shcEx1695 and shcEx1696[Pmyo-3::GFP], shcEx1697 and shcEx1698[Pttx-3::GFP], shcEx1699[Phlh-17::GFP].

Details on strains used in this study are listed in *Supplementary file 2*.

## EMS screen and mutant identification

To identify *cima-1* suppressors, animals that exhibited normal presynaptic distribution at the adult stage were isolated from a forward Ethyl Methane-Sulphonate (EMS) screen performed on the *cima-1(wy84)* mutants. The suppressor *ola226* was isolated from this screen. The causative genetic lesion was identified through SNP mapping and whole genome sequencing (*Minevich et al., 2012*) to be a G to A point mutation in the first exon of *mig-17*, turning E19 into K in the protein. Fosmid WRM0616aB07, which includes the *mig-17* gene, rescues the observed suppression of the AIY presynaptic distribution in *cima-1(wy84); ola226*.

## Germline transformation

Transformations were carried out by microinjection of plasmid DNA into the gonad of adult hermaphrodites (*Mello et al., 1991*). Plasmids were injected at 5–20 ng/μl concentrations.

## Plasmids

The following constructs were created by Gateway cloning (Invitrogen): P*mig-17::SL2::GFP*; P*mig-17::mig-17(E303A)::GFP*; P*hlh-17::mig-17*; P*unc-14::mig-17*; P*dpy-7::mig-17*; P*myo-3::mig-17*. The *mig-17* promoter is 1.7 kb sequence upstream from the start codon. The remaining constructs are listed in *Supplementary file 3*. Detailed cloning information is available upon request.

We constructed two Cas9-sgRNAs with pDD162 for each strain according to the method in *Dickinson et al. (2015)*. The repair template of *mig-17::mNeonGreen* was modified from pDD268 and is illustrated in *Figure 6—figure supplement 1B*. Briefly, *mNeonGreen* was flanked by 1.2 kb genomic sequence upstream or downstream of the *mig-17* stop codon. To prevent Cas9 from cutting the donor template, we also introduced one synonymous mutation in the protospacer adjacent motif (PAM). The repair template of *mig-17(E303A)* includes 1.2 kb upstream and 1.2 kb downstream of *mig-17* genomic sequence, which flank the Glutamic acid at the 303 site. We mutated the Glutamic acid (GAA) to Alanine (GCA) and introduced eight synonymous mutations to prevent Cas9 from cutting the donor template (*Figure 6—figure supplement 1A*). *mig-17(E303A)* point mutation or *mig-17::mNeonGreen* knock-in animals were generated by microinjection of 50 ng/μl Cas9-sgRNA plasmids, 20 ng/μl repair template, and 5 ng/μl P*myo-3::mCherry* as a co-injection marker. The engineered strains were screened by PCR and verified by Sanger sequencing. We examined the glia morphology and gonad defect in *mig-17::mNeonGreen* knock-in animals, and observe that they behave as wild type, suggesting that MIG-17::neonGreen does not compromise MIG-17 function.

## Protein extraction, digestion, and labeling

The samples were lysed in buffer (8 M guanidine hydrochloride, 100 mM TEAB) and sonicated. Samples were then centrifuged at 20,000 g for 30 min at 4°C, and the supernatant collected. Proteins were submitted to reduction by incubation with 10 mM DTT at 37°C for 45 min, followed by alkylation using 100 mM acrylamide for 1 hr at room temperature and digestion with Lys-C and trypsin using the FASP method (*Wiśniewski et al., 2009*). After stable isotope dimethyl labeling in 100 mM TEAB, peptides were mixed with light, intermediate and heavy (formaldehyde and NaBH3CN) isotopic reagents (1:1:1), respectively (*Boersema et al., 2009*). The peptide mixtures were desalted on a Poros R3 microcolumn according to the previous method (*Huang et al., 2018*).

## Liquid chromatography–tandem mass spectrometry (LC-MS/MS)

LC-ESI-MS/MS analyses were performed using an LTQ Orbitrap Elite mass spectrometer (Thermo Fisher Scientific, Bremen, Germany) coupled with a nanoflow EASY-nLC 1000 system (Thermo Fisher Scientific, Odense, Denmark). A two-column system was adopted for proteomic analysis. The mobile phases were in Solvent A (0.1% formic acid in $H_2O$) and Solvent B (0.1% formic acid in ACN). The derivatized peptides were eluted using the following gradients: 2–5% B in 2 min, 5–28% B in 98 min, 28–35% B in 5 min, 35–90% B in 2 min, 90% B for 13 min at a flow rate of 200 nl/min. Data-dependent analyses were used in MS analyses. The top 15 abundant ions in each MS scan were selected and fragmented in HCD mode.

Raw data was processed by Proteome Discover (Version 1.4, Thermo Fisher Scientific, Germany) and matched to the *C. elegans* database (20161228, 17,392 sequences) through the Mascot server (Version 2.3, Matrix Science, London, UK). Data was searched using the following parameters: 10 ppm mass tolerance for MS and 0.05 Da for MS/MS fragment ions; up to two missed cleavage sites were allowed; carbamidomethylation on cysteine, dimethyl labeling as fixed modifications; oxidation on methionine as variable modifications. The incorporated Target Decoy PSM Validator in Proteome Discoverer was used to validate the search results with only the hits with FDR $\leq$ 0.01. Three technical replicates were performed for the proteomic analyses.

## Microscopy and image analyses

Animals were anaesthetized with 50 mM Muscimol (Tocris) on 2% agarose pads (Biowest, Lot No.: 111860), and examined with either with Perkin Elmer or Andor Dragonfly Spinning-Disk Confocal Microscope Systems. Image processing was performed by using Volocity, Image J, Adobe Photoshop CS6 or Imaris software (Andor).

## Quantification

To quantify the percentage of animals with ectopic pre-synapses of AIY Zone one and posterior extension of glia, animals were synchronized by being selected at larva stage 4 (L4), and then examined 24 hr later using a Nikon Ni-U fluorescent microscope. Each dataset was collected from at least three biological replicates. At least 20 animals were scored for each group. For each germline transformation, multiple transgenic lines were examined. For synaptic allometric quantification, the ectopic synapses were defined as the presence of synaptic fluorescent markers the AIY Zone one region, an asynaptic area in wild type AIY neurons (*Colón-Ramos et al., 2007*; *Shao et al., 2013*). We also quantified the ratio of presynaptic length as the ratio of ventral length to total synaptic length (b/(a+b) in *Figure 2F*; *Shao et al., 2013*). The overlap of VCSC glia and ectopic synapses was defined as the VCSC glia and synaptic area of overlap at the Zone one and Zone two regions. The length of VCSC glial anterior process and ventral process (as shown in *Figure 3A*) were measured from confocal images taken in synchronized 1-day-old adults. The length of the pharynx and the body length were measured via DIC microscopy performed in synchronized 1-day-old adults.

The fluorescent intensity of MIG-17::mNeonGreen and EMB-9::mCherry in the pharyngeal region was normalized by the area with Image J from confocal images at the specified developmental stages. The mCherry clusters are likely intracellular accumulations of mCherry in the lysosome, as has been shown for other mCherry-tagged proteins. To minimize quantifying fluorescence from the intercellular EMB-9::mCherry clusters, we only quantified the mCherry in the second pharyngeal bulb region as shown in *Figure 7*.

## Electron microscopy

L4 animals were prepared for EM by high pressure freezing and freeze substitution as described (*Xuan et al., 2017*). Serial sections of 40 nm thickness cut on a Ultracut 7 (Leica) and collected on formvar-covered, carbon-coated copper grids (EMS, FCF2010-Cu), and post-stained with 2.5% uranyl acetate and lead citrate. Images were acquired on a FEI Tecnai G2 Spirit BioTWIN. AIY Zone 2 was identified based on anatomical landmarks at the base of the ventral nerve bundle (*White et al., 1986*).

## Statistical analysis

Specified statistical analyses were based on student's t-test for comparisons between two groups or one-way ANOVA by Tukey's multiple comparison test for three or more groups. All were analyzed using Prism 6.

## Acknowledgements

We thank ZF Altun and DH Hall from WormAtlas for help with schematic figures. We also thank Shiqing Cai, Yidong Shen, Kiyoji Nishiwaki, Mei Ding (Chinese Academy of Sciences), Yan Zou (Shanghai tech), David Sherwood (Duke University) and the Caenorhabditis Genetic Center (funded by NIH (P40 OD010440) for providing strains and plasmids. We thank members in Shao lab and the Colón-Ramos lab for insightful discussions on the work and advice on the project. We thank Mi Zhou for providing technical support on image acquisition. We thank the Yale CCMI Electron Microscopy Facility for use of their equipment. We thank the Research Center for Minority Institutions program, the Marine Biological Laboratories (MBL), and the Instituto de Neurobiología de la Universidad de Puerto Rico for providing meeting and brainstorming platforms. Research in the ZS lab was supported by the National Natural Science Foundation of China (31471026, 31872762), Shanghai Municipal Science and Technology Major Project (No. 2018SHZDZX01) and ZJLab. Research in the Zhang lab was supported by the National Natural Science Foundation of China (31870822). Research in the DAC-R lab was supported by NIH R01NS076558, DP1NS111778 and by an HHMI Scholar Award. We thank Life Science Editors for editing assistance.

## Additional information

### Funding

| Funder | Grant reference number | Author |
|---|---|---|
| National Natural Science Foundation of China | 31471026 | Jiale Fan<br>Tingting Ji<br>Kai Wang<br>Mengqing Wang<br>Xiaohua Dong<br>Yanjun Shi<br>Zhiyong Shao |
| NIH Office of the Director | DP1NS111778 | Laura Manning<br>Daniel A Colón-Ramos |
| National Institutes of Health | R01NS076558 | Laura Manning<br>Daniel A Colón-Ramos |
| Howard Hughes Medical Institute | Faculty Scholar | Daniel A Colón-Ramos |
| National Natural Science Foundation of China | 31872762 | Jiale Fan<br>Tingting Ji<br>Mengqing Wang<br>Xiaohua Dong<br>Yanjun Shi<br>Zhiyong Shao |
| Shanghai Municipal Science and Technology Major Project | 2018SHZDZX01 | Zhiyong Shao |
| National Natural Science Foundation of China | 31870822 | Jichang Huang<br>Xumin Zhang |

The funders had no role in study design, data collection and interpretation, or the decision to submit the work for publication.

### Author contributions

Jiale Fan, Xumin Zhang, Daniel A Colón-Ramos, Conceptualization, Resources, Funding acquisition, Investigation, Writing - original draft, Project administration, Writing - review and editing; Tingting Ji, Kai Wang, Jichang Huang, Mengqing Wang, Yanjun Shi, Investigation; Laura Manning,

Investigation, Writing - original draft; Xiaohua Dong, Zhiyong Shao, Conceptualization, Resources, Investigation, Writing - original draft, Project administration, Writing - review and editing

**Author ORCIDs**
Laura Manning (iD) https://orcid.org/0000-0003-1597-0600
Xumin Zhang (iD) https://orcid.org/0000-0002-2810-6363
Zhiyong Shao (iD) https://orcid.org/0000-0002-6475-7681
Daniel A Colón-Ramos (iD) https://orcid.org/0000-0003-0223-7717

**Decision letter and Author response**
Decision letter https://doi.org/10.7554/eLife.55890.sa1
Author response https://doi.org/10.7554/eLife.55890.sa2

## Additional files

### Supplementary files

• Supplementary file 1. Protein levels altered in *mig-17(ola226)* as detected by LS/MS proteomic analyses. Proteins upregulated (>1.2 fold) or downregulated (<0.8 fold) are listed in the spreadsheet. Note that *mig-17(ola226)* was isolated from forward genetic screen, which would introduce other background mutations. Further analyses are required to determine if the protein level changes are due specifically to the *mig-17(ola226)* mutation.

• Supplementary file 2. Strains used in this study List of strains and the corresponding genotypes used in this study.

• Supplementary file 3. Constructs used in this study. The name of constructs, the primers and the vectors (for building the constructs). Detailed cloning information is available upon request.

• Supplementary file 4. Key Resources Table.

• Transparent reporting form

### Data availability

All data is presented in the figures or supplementary figures.

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
