## [Decision Letter]

[Editors’ note: the authors submitted for reconsideration following the decision after peer review. What follows is the decision letter after the first round of review.]

Thank you for submitting your work entitled "ADAMTS-family MIG-17 regulates synaptic allometry by modifying extracellular matrix and glia morphology during growth" for consideration by *eLife*. Your article has been reviewed by three peer reviewers, and the evaluation has been overseen by a Reviewing Editor and a Senior Editor. The following individuals involved in review of your submission have agreed to reveal their identity: Andrew D Chisholm (Reviewer #2); Peri Kurshan (Reviewer #3).

Our decision has been reached after consultation between the reviewers. Based on these discussions and the individual reviews below, we regret to inform you that your work will not be considered further for publication in *eLife*.

Your manuscript addresses the important question of how synapses are maintained after embryonic development and how they are scaled during postembryonic growth (a process the authors refer to as allometry, which raised concerns with one of the reviewers). Using the nematode *C. elegans*, the authors had identified in a previous study the putative solute carrier transporter *cima-1* and the fibroblast growth factors as genes that function in the epidermis to shape glia, which are required to maintain the position of synapses of AIY interneurons. This current study extends these findings. The authors show that the *cima-1* loss of function defects in AIY synapse maintenance can be partially suppressed by loss of the ADAMTS metalloprotease *mig-17*. Overall, the topic is timely and important as our knowledge of synaptic maintenance is limited. However, the reviewers raised a number of concerns. The reviewers all agreed that a journal like Development or Genetics would be a perfect home for this story, but that the extent of novelty and mechanistic insight did not rise to what is expected from an *eLife* paper.

1) The extensive amount of previous work on *mig-17* in other cellular context somewhat diminished the novelty of this work. While *mig-17* is placed in a novel cellular context, the reviewers were not entirely convinced that this novel context is actually quite properly framed (is this really synaptic allometry? See below) or whether some of the findings are not merely a quite simple reflection of glial overgrowth which then resulted in several secondary phenomena (as discussed more below).

2) The authors propose that the *mig-17* ADAMTS enzyme negatively regulates the abundance of basement membrane proteins (such as collagen, which they show by both mass spec and fluorescence reporters). The interpretation is that this increase in BM proteins somehow suppresses defects caused by loss of the *cima-1* solute carrier protein, which itself negatively regulates the fibroblast growth factor in an adjacent tissue. How do all these molecules identified by the authors function together? Do the authors envision that *mig-17* is involved in the degradation of different types of basement membrane? Are the BM proteins direct or indirect targets? How would increased concentrations of BM proteins counteract the increased adhesion as a result of increased fibroblast growth factor coming from the epidermis that the authors propose in Shao et al.? Could *mig-17* have an effect on the fibroblast growth factor? Do some of these factors physically interact? Some of these questions are obviously beyond the scope of this manuscript, but we would have liked to see somewhat more mechanistic insight into this important and interesting process.

3) The picture emerging from the author's analysis of genetic interactions is not entirely clear. The set of relationships between many different molecular players is very hard to keep straight, and the authors make sometimes contradictory statements about those relationships. For example, the authors state that the different molecules "act in the same pathway", which is confusing because they later explain (and their model figure shows) that there are two separate pathways (one from epidermal cells and one from muscle) that both converge on glial outgrowth. As another example, their explanation for how basement membrane components such as collagen (EMB-9) factor in to the pathway are confusing, perhaps owing to the fact that the alleles they use are not completely well-characterized. But a careful piecing together of all the data and genetic interactions suggests that an over-abundance (or extra stable version) of EMB-9 seems to suppress glial overgrowth (and therefore ectopic synapse formation). Their model figure uses arrow to show that relationships exist between MIG-17 and basement membrane components such as EMB-9 and glial cells, but a more accurate model would show that MIG-17 negatively regulates EMB-9, which in turn negatively regulates glial over-growth. Which is why either a loss-of-function *mig-17* allele or a gain-of-function *emb-9* allele, which both would lead to an increase in EMB-9 levels, suppress the *cima-1* phenotype.

4) Extending on the points above: The *mig-17* secreted protease likely functions from muscle to suppress the *cima-1* mutant defects and requires enzymatic activity. Thus, different tissues (epidermis and muscle) are involved in maintaining glia shape, which in turn is necessary to maintain AIY synapses. The authors further show by mass spectrometry and fluorescent reporters that the abundance of basement membrane (BM) proteins is increased in *mig-17* mutants. This thorough genetics study clearly establishes a role for *mig-17* in the process of synapse maintenance, which is distinct from the role of *mig-17* in shaping gonad development. Their data suggests that BM proteins such as collagens are involved in the process by which *mig-17* functions to establish and maintain glial shape. Conceptually, however, the idea that *mig-17* remodels the basement membrane is not novel (even if there are differences in molecular mechanism from e.g. gonad development). For example, *mig-17* catalytic activity is required for function in other contexts (Nishiwaki et al., 2000, using the same missense mutation as used here). Other details (*mig-17* effects on organ size; *mig-17* reporter pattern, developmental regulation of *mig-17* and *emb-9*) also cover similar ground to that reported by others (as properly acknowledged by the authors).

5) The authors frame their work in the context of “synaptic allometry” (maintenance of synaptic position during growth) yet it remains unclear if the ectopic synapses of *cima-1* mutants result from disrupted allometry, or are a more specific result of the abnormal glial morphology. For example, most of the genetic manipulations appear to have little effect on synaptic allometry of “normal” (non-ectopic) synapses. Taken together, "allometry" may be an over-statement, as the phenotype can be boiled down to glial overgrowth (and resulting ectopic presynaptic puncta formation). The main novelty seems to be that both muscle and epidermal-derived signals impinge on glia morphology, but I'm not sure if that rises to the level of significance required.

6) There are also a number of additional concerns that seem minor in isolation, but together add up to the level of substantial concerns:

a) The statistical analyses used are not appropriate – the authors use t-tests for every individual comparison within a group, when they instead should use an ANOVA for multiple comparisons between the members of the group. Some of their statistical significance may erode upon this more stringent analysis.

b) The localization pattern of EMB-9::mCherry does not look like what you would expect from something that is supposed to be encasing muscles… what are all the clusters? Are they overexpression artifacts? mCherry aggregation artifacts? How was the quantification done and did it include the clusters? The colocalization between MIG-17::neonGreen and EMB-9::mCherry is not convincing.

c) They use Punc-14 promoter for ubiquitous neuronal expression is not standard protocol anymore. There is evidence for ubiquitous expression of this promoter.

[Editors’ note: further revisions were suggested prior to acceptance, as described below.]

Thank you for submitting your article "A muscle-epidermis-glia signaling axis sustains synaptic specificity during allometric growth in *C. elegans*" for consideration by *eLife*. Your article has been reviewed by two of the three peer reviewers that had seen an earlier version of the manuscript that was rejected, an additional new reviewer, and the evaluation has been overseen by a Reviewing Editor and Didier Stainier as the Senior Editor. The following individual involved in review of your submission has agreed to reveal their identity: Peri Kurshan (Reviewer #1).

The two original reviewers have very much appreciated the revision of the manuscript and they agree (together with the new reviewer) that this manuscript now properly emphasizes an important conceptual advance by (a) framing the problem along the allometry concept, and by (b) demonstrating complex interactions between different tissues types that ensure allometry. However, one important experiment remains to be done (point #4 by reviewer #2) that we agreed would be necessary to strengthen an important, and somewhat understated (see reviewer #1) premise of the paper, namely that the effect of the ECM on synapse maintenance is indirect. Together with other points in the manuscript, we consider this to be an important novelty that would make the manuscript of interest to *eLife*. As per further discussion between the reviewers, this point should not only be addressed by examining the *emb-9* marker, but also by a more comprehensive analysis of available EM sections.

All other comments by reviewer #2 can be editorially addressed.

Reviewer #1:

I find this version of the paper significantly more compelling.

The authors now emphasize that they have uncovered a signaling cascade mediating synapse position during growth that begins in the epidermis and muscle, goes through the glia, and ends at the neuron. I do think this is an interesting finding, and the paper now does a good job of laying out the novelty.

The question in my mind is how translatable this is to other systems. On the one hand, *C. elegans* as a model system has the advantage of enabling the authors to work out a complex genetic cascade such as this one within its in vivo context. But on the other hand, we already knew that glia are important for dictating where synapses form. In this case, the glia are getting signals from other structural cells (e.g. epidermis). But that is not going to be a widespread mechanism in the mammalian CNS… the authors would have been better served by focusing on the fact that the extracellular matrix is dictating glia position and thereby synapse position (rather than emphasizing the source of those signals as they do with the title of the paper)… at least that is something that could be directly relevant in other systems.

So in my opinion, what is most interesting is that ECM structure and composition can dictate synapse position not only directly, as has been shown before, but also indirectly by "repositioning" glia, as shown here. That's a slightly different spin than the one they've put on this.

The authors still have not really figured out the mechanisms by which this process occurs, and it is confusing that both putative lof and gof alleles of collagen seem to have the same effect on glial positioning.

Reviewer #2:

This manuscript by Fan and colleagues is a revised version of an earlier manuscript. It addresses the important question of how synapses are maintained in relation to other tissues after embryonic development and how they are scaled during postembryonic growth (a process the authors term allometry). Using the nematode *C. elegans*, the authors had previously identified the putative solute carrier transporter *cima-1* and the fibroblast growth factors as genes that function in the epidermis to shape glia, which in turn are required to maintain the position of synapses of AIY interneurons. This current study extends these findings. Using a forward genetic screen, the authors identify a genetic suppressor of the *cima-1* loss of function phenotype. This suppressor turns out to be *mig-17*, an ADAMTS protease, which is secreted to directly or indirectly remodel the basement membrane between the muscle and the glial cell. Through masspectrometry, the authors identify basement membrane proteins that are upregulated in *mig-17* mutants. Removing some of the upregulated basement membrane proteins also suppresses the *cima-1* loss of function phenotype suggesting that defective basement membranes are responsible for the suppression. Overall, the genetic results are clear and support the model the authors propose. On the downside, some of the genetic effects are possibly rather unspecific. For example, both gain and loss of function mutations of the collagen *emb-9* suppress the *cima-1* phenotype, suggesting that general basement membrane “malaise” is sufficient for the observed suppression. The authors acknowledge this, if not in these words. Along those lines, the genes that are mediating synaptic allometry as part of common genetic pathways in different tissues have, with some exceptions, little mechanistic connection, leaving many of the molecular details of this interesting phenomenon in the dark. That all being said, the finding that the glial cells (which mediate the allometry), are not only regulated by the adjacent epidermis, but also are influenced by a basement membrane that is secreted (or remodeled) by another adjacent tissue, namely muscle, is an important extension of the concept of how the AIY synapses are maintained. The technical and conceptual criticisms of the initial version have by and large been addressed and the writing is much improved, making the concepts much clearer.

Reviewer #3:

This solid and important manuscript reports on cellular and molecular bases of sustained synapse position in the face of organismal allometric growth. The authors identify a cooperative mechanism between non-neuronal cells (epidermis, BM-secreting muscle cells, and glia), whereby maintained glia morphology/position leads to the correct positioning of presynaptic specializations. Whereas the authors link *mig-17* and BM components to this mechanism, no precise mechanistic/molecular understanding is provided, diminishing the significance of the study, especially in view of previously described work on *mig-17* (which the authors cite), and the lab's previous work with *cima-1* and *egl-15*. However, important conceptual novelty arises from the authors' identification of a multipartite system in which secreted molecules from distinct classes of non-neuronal cells coordinately impact glia position, thereby influencing synaptic development. Such multicellular coordination driving synaptic development may be at play in other species as well. The resolution enabled by *C. elegans* reveals an important mechanistic principle.

1) The use of the term "synapse allometry" may be an overstatement in view of the fact that only presynaptic specializations are affected/examined here. The position of these presynaptic structures appears hard wired, based solely on the region of contact between glia and presynaptic neurite, which is very interesting. But can these be considered bona fide synapses?

2) To further highlight the importance of non-neuronal cells in synapse positioning/development, the authors could/should underscore the fact that the post-synaptic partner (RIA) has no impact on presynaptic specialization assembly in the Discussion.

3) An allele of *mig-17, ola226*, was isolated as a suppressor mutation of *cima-1* mutants, which otherwise display altered VCSC glia morphology and concomitant ectopic presynaptic specializations. Rescue assays for *mig-17* function are logically carried out in double mutants *cima-1;mig-17*, testing whether transgenic animals are rescued back to *cima-1* mutant phenotype, i.e. now displaying ectopic synapses. Since "rescue" in this situation is the manifestation of a defective phenotype, the interpretation provided would much strengthened if transgenic expression of *mig-17* in the WT background showed no ectopic synapses.

4) Based on White's electron micrographs, the authors write that "No BM is present between the VCSC and the nerve ring neurons". While the image presented does appear to show this, BM on EM may not always be so clearly visible, especially if cells are densely apposed with each other. The authors could use confocal microscopy with fluorescent reporters for VCSC glia and EMB-9::mCherry, as well as for the AIY neuron and EMB-9::mCherry, to further validate this point. This is important as much of the novelty of this work relies on BM not directly impacting these synapses, and rather an indirect effect of BM on synapse position via glia.

5) The authors write that EMB-9::mCherry accumulations "are likely intracellular/lysosomal" and are therefore excluded from the analysis. While this may be plausible, colocalization with a lysosomal marker would enable a clearer interpretation.

6) Authors say that *ola226* mutants have abnormal glia morphology and that AIY neurite and soma are anteriorly positioned. Anteriorly positioned with respect to what? Using the pharynx as a reference, as it appears from the schemtics presented, when the mutants under study affect pharynx size, may be problematic. How positions are evaluated should be described.

---

## [Author Response]

[Editors’ note: the authors resubmitted a revised version of the paper for consideration. What follows is the authors’ response to the first round of review.]

Your manuscript addresses the important question of how synapses are maintained after embryonic development and how they are scaled during postembryonic growth (a process the authors refer to as allometry, which raised concerns with one of the reviewers). Using the nematode *C. elegans* the authors had identified in a previous study the putative solute carrier transporter cima-1 and the fibroblast growth factors as genes that function in the epidermis to shape glia, which are required to maintain the position of synapses of AIY interneurons. This current study extends these findings. The authors show that the cima-1 loss of function defects in AIY synapse maintenance can be partially suppressed by loss of the ADAMTS metalloprotease mig-17. Overall, the topic is timely and important as our knowledge of synaptic maintenance is limited. However, the reviewers raised a number of concerns. The reviewers all agreed that a journal like Development or Genetics would be a perfect home for this story, but that the extent of novelty and mechanistic insight did not rise to what is expected from an eLife paper.

We received the constructive comments from the reviewers, and based on their thoughtful suggestions re-wrote the paper to better highlight the significance of the study. We also integrated all their other suggestions, including analyses of new alleles, statistical methods, etc. While the manuscript contains similar data as the original submission (with the suggested additions from the reviewers), we believe the reframing and rewriting improves it significantly to better underscore its main findings.

Briefly, we agree with the reviewers that the role of MIG-17 in remodeling the basement membrane is not a new finding. Upon rereading the critiques and the manuscript, we also agree that in our detailed genetic analyses we unintentionally placed too much emphasis on the molecules that emerged from our proteomic and genetic screens, drawing away from the important concepts that we learned. We thank the reviewers for pointing this out and helping us more clearly frame the value of this study.

The main finding here, which we now emphasize by re-writing the manuscript, is that a muscle-epidermis-glia signaling axis sustains synaptic positions during growth in *C. elegans*. The contribution of the paper is on the concept of synaptic allometry, and the role of non-neuronal cells, like muscles and the epidermal cells, in collaborating towards the maintenance of glia position, which in turn sustains synaptic positions during growth. I have presented this work in several forums in which it has been well received, particularly by the scientific community which work on glia and the role of non-neuronal cells in regulating nervous system architecture and function. The rigor with which we examine the contribution of specific tissues in positioning synapses in vivo is not possible in most other systems, and the important conceptual lessons which emerge from the study help inform examination of the process of synaptic allometry in other contexts.

1) The extensive amount of previous work on mig-17 in other cellular context somewhat diminished the enthusiasm of the novelty of this work. While mig-17 is placed in a novel cellular context, the reviewers were not entirely convinced that this novel context is actually quite properly framed (is this really synaptic allometry? See below) or whether some of the findings are not merely a quite simple reflection of glial overgrowth which then resulted in several secondary phenomena (as discussed more below).

We have rewritten the paper to better explain its conceptual contributions.

Briefly, the concept emerging from this study, beyond MIG-17, is that the circuit architecture established during embryogenesis is maintained during growth through interactions with non-neuronal cells, particularly muscles and epidermal cells. We find that muscles regulate this process through the modulation of the basement membrane via MIG-17. To be sure, the role of ADAMTS family protein GON-1 in synapse morphology has been described for NMJs. Yet, an important difference between those synapses and the ones studied here is that the basement membrane in NMJs is in direct contact, and likely directly signals, to NMJ synapses. In this study, the examined synapses in the nerve ring are not in direct contact with the basement membrane. So then how is it that the BM helps sustains synaptic positions during growth? We found the answer is by modifying, not directly the synapse, but glia position and morphology, which in turn positions the synapses during growth.

So, as the reviewers correctly point out, the synaptic phenotype emerges from the changing of glial morphology. We do not see this as a trivial finding. Based on our experience publishing in glia relationship with synapses, I can confidently state that there are very few studies linking glia to synaptic positions in vivo, and to my knowledge, no studies linking signaling from non-neuronal cells and the BM in positioning glia during growth to then regulate synaptic positions. Given that the relationship we examine here between BM:glia:synapses is also present in mammalian CNS systems (including the blood-brain barrier), and given that sustaining synaptic positions during growth (which we term synaptic allometry) is a conserved principle, we believe the concepts learned here might have important implications for our understanding of how synaptic positions are sustained during growth, and the role of non-neuronal cells, such as glia, muscles and epidermal cells, in forming a signaling axis that links growth information with the scaling of the synaptic pattern to preserve embryonically-derived circuit architecture.

2) The authors propose that the mig-17 ADAMTS enzyme negatively regulates the abundance of basement membrane proteins (such as collagen, which they show by both mass spec and fluorescence reporters). The interpretation is that this increase in BM proteins somehow suppresses defects caused by loss of the cima-1 solute carrier protein, which itself negatively regulates the fibroblast growth factor in an adjacent tissue. How do all these molecules identified by the authors function together? Do the authors envision that mig-17 is involved in the degradation of different types of basement membrane? Are the BM proteins direct or indirect targets? How would increased concentrations of BM proteins counteract the increased adhesion as a result of increased fibroblast growth factor coming from the epidermis that the authors propose in Shao et al.? Could mig-17 have an effect on the fibroblast growth factor? Do some of these factors physically interact? Some of these questions are obviously beyond the scope of this manuscript, but we would have liked to see somewhat more mechanistic insight into this important and interesting process.

We have rewritten these sections to address these points.

Briefly, our model is that epidermal cells establish an adhesion with glia through the FGF receptor. That adhesion is negatively regulated during growth by CIMA-1. Reduction of CIMA-1 will result in more adhesion and the posterior displacement of the glia. But the displacement of the glia requires modification of the basement membrane by MIG-17. When the basement membrane is not properly modified, like in BM mutants, or in mig-17 mutants, the glia can’t be posteriorly displaced and the morphological phenotypes (and ectopic synapses) are suppressed.

We explain this now much more clearly when discussing, for instance, the different alleles tested for *emb-9*, and for *emb-9*- regulating proteins. While these alleles are known to have different effects on the conformation or levels of the EMB-9 protein in the basement membrane, they all suppress the synaptic phenotype in *cima-1(wy84)* mutants. Their shared ability to suppress *cima- 1(wy84)* mutants suggests that lesions resulting in defects in the basement membrane prevent the repositioning of glia that gives rise to the ectopic synapses. Together with our genetic and proteomic findings, they support a model whereby MIG-17 regulates basement membrane proteins, such as EMB-9, to modulate glia position during growth, and synaptic allometry.

While we did not overexpress BM proteins, overexpression of MIG-17 results in glia extension and phenotypes like CIMA-1. The synaptic allometric defect of EGL-15A overexpression also depends on MIG-17. We cannot exclude a role of MIG-17 in regulating the FGF receptor (which we now mention in the paper), we found that MIG-17 is epistatic to the FGF receptor, suggesting that *mig-17* acts downstream (or in parallel) to modulate the role of *elg-15* in positioning glia and regulating synaptic allometry. We better discuss these findings in the Results and the Discussion.

3) The picture emerging from the author's analysis of genetic interactions is not entirely clear. The set of relationships between many different molecular players is very hard to keep straight, and the authors make sometimes contradictory statements about those relationships. For example, the authors state that the different molecules "act in the same pathway", which is confusing because they later explain (and their model figure shows) that there are two separate pathways (one from epidermal cells and one from muscle) that both converge on glial outgrowth. As another example, their explanation for how basement membrane components such as collagen (EMB-9) factor in to the pathway are confusing, perhaps owing to the fact that the alleles they use are not completely well-characterized. But a careful piecing together of all the data and genetic interactions suggests that an over-abundance (or extra stable version) of EMB-9 seems to suppress glial overgrowth (and therefore ectopic synapse formation). Their model figure uses arrow to show that relationships exist between MIG-17 and basement membrane components such as EMB-9 and glial cells, but a more accurate model would show that MIG-17 negatively regulates EMB-9, which in turn negatively regulates glial over-growth. Which is why either a loss-of-function mig-17 allele or a gain-of-function emb-9 allele, which both would lead to an increase in EMB-9 levels, suppress the cima-1 phenotype.

We thank the reviewers for this comment. While the genetic interactions are consistent with our model, we agree that the large number of double and triple mutants in the examination of our genotypes make it both hard to read and unclear. We addressed this in two ways:

1) We add schematics to underscore the pathways we found and the tissues in which they act. Briefly, when we (confusingly) referred to a single pathway, we meant it in a genetic sense – a genetic pathway culminating in synaptic allometry. We clarify this and other points raised by the reviewer sin the text and figures.

2) We re-wrote the text with professional editorial help to better explain the phenotypes in the context of the examined mutants.

4) Extending on the points above: The mig-17 secreted protease likely functions from muscle to suppress the cima-1 mutant defects and requires enzymatic activity. Thus, different tissues (epidermis and muscle) are involved in maintaining glia shape, which in turn is necessary to maintain AIY synapses. The authors further show by mass spectrometry and fluorescent reporters that the abundance of basement membrane (BM) proteins is increased in mig-17 mutants. This thorough genetics study clearly establishes a role for mig-17 in the process of synapse maintenance, which is distinct from the role of mig-17 in shaping gonad development. Their data suggests that BM proteins such as collagens are involved in the process by which mig-17 functions to establish and maintain glial shape. Conceptually, however, the idea that mig-17 remodels the basement membrane is not novel (even if there are differences in molecular mechanism from e.g. gonad development). For example, mig-17 catalytic activity is required for function in other contexts (Nishiwaki et al., 2000, using the same missense mutation as used here). Other details (mig-17 effects on organ size; mig-17 reporter pattern, developmental regulation of mig-17 and emb-9) also cover similar ground to that reported by others (as properly acknowledged by the authors).

We agree that, while we go into a rigorous dissection of the proteomic and genetic findings to establish the link between MIG-17 and the BM, that is not the main novelty of the study. The novelty is that through the reorganization of the BM, MIG-17 cooperates with signals from epidermal cells to maintain glia position, and in that way modulate synaptic positions during growth (as explained in more detail in point #1 above).

5) The authors frame their work in the context of “synaptic allometry” (maintenance of synaptic position during growth) yet it remains unclear if the ectopic synapses of cima-1 mutants result from disrupted allometry, or are a more specific result of the abnormal glial morphology. For example, most of the genetic manipulations appear to have little effect on synaptic allometry of “normal” (non-ectopic) synapses. Taken together, "allometry" may be an over-statement, as the phenotype can be boiled down to glial overgrowth (and resulting ectopic presynaptic puncta formation). The main novelty seems to be that both muscle and epidermal-derived signals impinge on glia morphology, but I'm not sure if that rises to the level of significance required.

This is an important point which we better addressed in the revised manuscript:

A) In this manuscript we introduce the term “synaptic allometry”, an important concept, in our opinion, for the community. As we define it here, synaptic allometry refers to sustaining the synaptic pattern during growth. This is different than maintaining the morphology/position/function/protein composition of one synapse. It is a nuanced, but important point. We would argue that sustaining the synaptic positions is what is important to sustain the synaptic pattern that underlies the connectivity. We now re-wrote the text to define clearly what we mean by synaptic allometry.

B) Extending on this point, we now show in the supplementary figures the relationship between the ectopic synapses in the *cima-1* mutants and the postsynaptic partner RIA, demonstrating that the change in the presynaptic pattern due to glia morphology affects the relative positions of the components of the synapse, and the connectivity.

C) As the reviewers point out, the synaptic phenotype emerges from the changes in glial morphology. We do not see this as a trivial finding. There are no studies to our knowledge linking signalling from non-neuronal cells and the BM in positioning glia during growth to then regulate synaptic positions. The role of nonneuronal cells, in vivo, in sustaining synaptic positions is an important question which we address here. How glia morphology is maintained, and how this maintenance then impinges on the circuit architecture is essentially unknown and an important question in neuroscience. We find here that the glia mediate signals from epidermal cells and the BM regulated by muscles, to maintain synaptic positions during growth.

6) There are also a number of additional concerns that seem minor in isolation, but together add up to the level of substantial concerns:a) The statistical analyses used are not appropriate – the authors use t-tests for every individual comparison within a group, when they instead should use an ANOVA for multiple comparisons between the members of the group. Some of their statistical significance may erode upon this more stringent analysis.

Thanks for the comments. We now use the ANOVA analysis for comparison among groups of three or more. None of the conclusions changed, but this is an important point and we are thankful for the correction.

b) The localization pattern of EMB-9::mCherry does not look like what you would expect from something that is supposed to be encasing muscles… what are all the clusters? Are they overexpression artifacts? mCherry aggregation artifacts? How was the quantification done and did it include the clusters? The colocalization between MIG-17::neonGreen and EMB-9::mCherry is not convincing.

The clusters are likely mCherry accumulations in lysosomes. While GFP gets quenched in acidic environments, mCherry does not (we carefully characterized this for a different protein and its relationship to the autophagy pathway in (Hill et al., 2019)). For this study, and to avoid quantifying these intracellular lysosomal aggregates, we carefully quantified the fluorescent intensity in the region of second pharyngeal bulb without scoring the fluorescence from the aggregates. This is now indicated both in the figures and the Materials and methods.

c) They use Punc-14 promoter for ubiquitous neuronal expression is not standard protocol anymore. There is evidence for ubiquitous expression of this promoter.

This is a good point. We redid these experiments with a number of additional promoters, and now demonstrate that, consistent with *mig-17* being a secreted protein, expression from any cell-specific promoter rescues the phenotype (including the pan-neuronal *rab-3* promoter, nerve ring specific *lim-4* promoter and AIY specific *ttx-3* promoter. These data are now included in Figure 5—figure supplement 1). Note that this findings are different from the negative result we obtained with the *unc-14* promoter, and we revisit our interpretation based on these more rigorous experiments. While they do not change the overall conclusion, they extend our understanding of *mig-17* as a secreted protein, and we present that accordingly.

[Editors’ note: what follows is the authors’ response to the second round of review.]

The two original reviewers have very much appreciated the revision of the manuscript and they agree (together with the new reviewer) that this manuscript now properly emphasizes an important conceptual advance by (a) framing the problem along the allometry concept, and by (b) demonstrating complex interactions between different tissues types that ensure allometry. However, one important experiment remains to be done (point #4 by reviewer #2) that we agreed would be necessary to strengthen an important, and somewhat understated (see reviewer #1) premise of the paper, namely that the effect of the ECM on synapse maintenance is indirect. Together with other points in the manuscript, we consider this to be an important novelty that would make the manuscript of interest to eLife. As per further discussion between the reviewers, this point should not only be addressed by examining the emb-9 marker, but also by a more comprehensive analysis of available EM sections.All other comments by reviewer #2 can be editorially addressed.

Please find our revised manuscript “A muscle-epidermis-glia signaling axis sustains synaptic positions during growth in *C. elegans*”. We integrated most of the reviewer’s comments, as suggested. In particular, we have now added additional data demonstrating the relationship between the basement membrane and the nerve ring (new Figure 9—figure supplement 1). We do this by showing the relative position of EMB-9 marker with a body muscle, with glia and with AIY marker, and demonstrate by fluorescence microscopy that EMB-9 localizes primarily to muscles (pharynx and body wall muscles), near the glia, but separate from the neuron (even AIY, which is near the edge of the nerve ring). Notably, in the images one can observe a darker area where the nerve ring is located, indicative of reduced of EMB-9 labeling. We also performed electron microscopy in AIY. Although it is only a panel in the Figure 9—figure supplement 1, we are very proud of this achievement, as it is technically challenging to identify the AIY neuron in the context of the nerve ring. We carefully inspected new electron micrographs we generated, and, consistent with John White’s published electron micrographs, we observe that the basement membrane near the nerve ring is only visible in the region abutting the pseudocoelum. We confirmed our observations by communicating directly with EM expert David Hall (personal communication), who also reported to us that in the hundreds of electron micrographs he has reconstructed and inspected, he does not observe basement membrane in the nerve ring. Our findings are consistent with the observations made in other systems that the basement membrane is an organized extracellular matrix structure that is 30-70 nanometers thick, and not present in the fascicles of the CNS (Heikkinen et al., 2014; Krishnaswamy et al., 2019). We make the distinction in the figure legend, however, that while the *structure* of the basement membrane might not be present inside the nerve ring, neurons in the nerve ring could express basement membrane proteins.